



# Estimating the turbulent kinetic energy dissipation rate from one-dimensional velocity measurements in time

Marcel Schröder[1,2], Tobias Bätge[1,2], Eberhard Bodenschatz[1,2,3], Michael Wilczek[1,4], and Gholamhossein Bagheri[1]

[1]Max Planck Institute for Dynamics and Self-Organization (MPIDS), Am Faßberg 17, 37077 Göttingen, Germany
[2]Faculty of Physics, Georg August University Göttingen, Friedrich-Hund-Platz 1, 37077 Göttingen, Germany
[3]Physics Department, Cornell University, 523 Clark Hall, Ithaca, NY, USA
[4]Theoretical Physics I, University of Bayreuth, Universitätsstr. 30, 95447 Bayreuth, Germany

**Correspondence:** Gholamhossein Bagheri (gholamhossein.bagheri@ds.mpg.de)

**Abstract.** The turbulent kinetic energy dissipation rate is one of the most important quantities characterizing turbulence. Experimental studies of a turbulent flow in terms of the energy dissipation rate often rely on one-dimensional measurements of the flow velocity fluctuations in time. In this work, we first use Direct Numerical Simulation (DNS) of Stationary Homogeneous Isotropic (SHI) turbulence at Taylor-scale Reynolds numbers $74 \leq R_\lambda \leq 321$ to evaluate different methods for inferring the energy dissipation rate from one-dimensional velocity time records. We systematically investigate the influence of the finite turbulence intensity and the misalignment between the mean flow direction and the measurement probe, and derive analytical expressions for the errors associated with these parameters. We further investigate how statistical averaging for different time windows affects the results as a function of $R_\lambda$. The results are then combined with Max Planck Variable Density Turbulence Tunnel (VDTT) hot-wire measurements at $147 \leq R_\lambda \leq 5864$ to investigate flow conditions similar to those in the atmospheric boundary layer.

## 1 Introduction

Turbulence is fundamental to many natural and engineering processes, such as transport of heat and moisture in the Earth's atmosphere (e.g. Wyngaard, 1992; Garratt, 1994; Muschinski and Lenschow, 2001; Fairall and Larsen, 1986; Hsieh and Katul, 1997), wind energy conversion (Smalikho et al., 2013), entrainment and mixing (e.g. Warhaft, 2000; Sreenivasan, 2004; Deshpande et al., 2009; Gerber et al., 2008, 2013; Siebert et al., 2013; Fodor and Mellado, 2020), and warm rain initiation (e.g. Shaw, 2003; Devenish et al., 2012; Pumir and Wilkinson, 2016; Li et al., 2020), to name just a few. In three-dimensional turbulence, the kinetic energy is typically injected into the flow at the largest scales and successively transferred to smaller eddies by means of the direct energy cascade. At the smallest scales characterized by the Kolmogorov length scale (or the dissipation scale) $\eta_K$, kinetic energy is dissipated by viscous effects at the energy dissipation rate $\epsilon$. The energy dissipation rate $\epsilon$ is one of the most fundamental quantities in turbulence and is used to estimate many relevant features of a turbulent flow, such as the Kolmogorov length scale $\eta_K$, the Taylor microscale $\lambda$, the Taylor-scale Reynolds number $R_\lambda$ and, by means of dimensional estimates, the energy injection scale.



The instantaneous energy dissipation field $\epsilon_0(\boldsymbol{x}, t)$ is highly intermittent with strong small-scale fluctuations (Pope, 2000; Davidson, 2015, and references therein), which are at the core of the intermittency problem in turbulence (Sreenivasan and An-
tonia, 1997; Muschinski et al., 2004; Buaria et al., 2019). It also plays an important role in turbulent mixing in reacting flows (e.g. Sreenivasan, 2004; Hamlington et al., 2012; Sreenivasan, 2019) or turbulence-induced rain initiation in warm clouds (Devenish et al., 2012). $\epsilon_0(\boldsymbol{x}, t)$, however, is extremely difficult to measure experimentally as it requires the complete knowledge of the three-dimensional velocity field at high temporal and spatial resolution.

Apart from the instantaneous dissipation field $\epsilon_0(\boldsymbol{x}, t)$, the energy dissipation in a turbulent flow can be statistically described
by either the local or global mean energy dissipation rate, which are both important. Local volume averages of the instantaneous dissipation field $\langle \epsilon \rangle_R$ and related surrogates, e.g. longitudinal, transverse or off-diagonal components of $\epsilon_0(\boldsymbol{x}, t)$, still capture intermittent effects of turbulence (Lefeuvre et al., 2014; Almalkie and de Bruyn Kops, 2012, and references therein). The local volume averages of the dissipation field converge to the global mean energy dissipation rate $\langle \epsilon \rangle$ for sufficiently large averaging volumes. $\langle \epsilon \rangle$ can be used to describe the statistics of homogeneous and locally isotropic turbulence based on the Kolmogorov's
phenomenology (K41) (Kolmogorov, 1941). However, even if the global mean energy dissipation rate $\langle \epsilon \rangle$ is known with a low uncertainty, it is of great value to know how locally averaged dissipation rates $\langle \epsilon \rangle_R$ deviate from the global mean energy dissipation rate $\langle \epsilon \rangle$.

For a statistically stationary homogeneous isotropic (SHI) turbulent flow, $\langle \epsilon \rangle$ can be estimated from time-dependent single-point one-dimensional velocity measurements through different methods, such as longitudinal or transverse velocity gradients
(Wyngaard and Clifford, 1977; Elsner and Elsner, 1996; Antonia, 2003; Siebert et al., 2006, among others), inertial-range scaling laws comprising the famous 4/5 law (Kolmogorov, 1941, 1991), counting zero-crossings of the velocity fluctuation time series (Sreenivasan et al., 1983; Wacławczyk et al., 2017) or dimensional arguments (e.g. Taylor, 1935; McComb et al., 2010; Vassilicos, 2015). These methods usually invoke Taylor's hypothesis to map temporal signals onto spatial signals, which requires a *sufficiently small* turbulence intensity. The turbulence intensity is defined as the ratio of the root mean square velocity
fluctuations $\sigma_{u'}$ to the mean velocity $U$. When all of these criteria are met, single-point velocity measurements with hot-wire anemometers at a high temporal resolution have been shown to be suitable for accurately estimating the global energy dissipation rate (Lewis et al., 2021; Sinhuber, 2015; Elsner and Elsner, 1996; Antonia, 2003). However, *ideal* SHI and low-intensity turbulent flows are rarely encountered in natural turbulent flows, such as those in the atmospheric boundary layer.

In such non-stationary and inhomogeneous flows, the global mean energy dissipation rate $\langle \epsilon \rangle$ alone is not representative as
the characteristics of turbulent flows can be highly time- and space-dependent even at the energy injection scales. As a result, one needs to calculate a local $\langle \epsilon \rangle_\tau$ and $\langle \epsilon \rangle_R$, respectively, based on velocity statistics for a *properly chosen* averaging window $\tau$ in time or $R$ in space, which is short enough for resolving the temporal or spatial variations but also long enough to obtain statistically representative values with *acceptable* systematic and/or random errors (e.g. Wyngaard, 1992; Lenschow et al., 1994). Therefore, a conflict arises with respect to the averaging time between resolving small-scale features of a turbulent flow
and statistical convergence under non-stationary and inhomogeneous conditions.

In the case of atmospheric flows, in *situ* measurements made via airborne (Malinowski et al., 2013; Siebert et al., 2006, 2013; Muschinski et al., 2004; Frehlich et al., 2004; Nowak et al., 2021; Dodson and Small Griswold, 2021, e.g.) as well as ground-





based (Chamecki and Dias, 2004; O'Connor et al., 2010; Risius et al., 2015; Siebert et al., 2015, e.g.) platforms typically can only resolve the coarse-grained time series of the local mean energy dissipation rate $\langle\epsilon\rangle_\tau$. However, since there is no high-resolution three-dimensional velocity measurement available during such in *situ* measurements to serve as the ground-truth, it remains unclear how large the errors in estimating the coarse-grained time series of the local mean energy dissipation rate are due to individual choices of the averaging window. In the absence of a ground-truth reference, the comparison between different methods exposes large deviations (Wacławczyk et al., 2020; Siebert et al., 2006; Risius et al., 2015; Wacławczyk et al., 2017). As an example, Wacławczyk et al. (2020) found deviations of about 5%-50% for estimating the mean energy dissipation rate depending on the method and averaging windows using synthetic data modeled via a von Karman spectrum. Another example is the work of Akinlabi et al. (2019), who found that estimates of mean energy dissipation rate by one-dimensional longitudinal velocity can differ by a factor of 2 to 3 from those calculated using DNS, depending on the method used.

Our literature review indicates that a systematic investigation is still needed to fully understand how the choice of averaging window, analysis methods, turbulence intensity and large-scale random flow velocities can influence estimating the mean energy dissipation rate and its deviations from the instantaneous energy dissipation rate. To this end, we systematically benchmark different techniques available in the literature using fully resolved DNS of statistically stationary, homogeneous, isotropic turbulence. Since the full dissipation field is available from DNS, this approach provides ground-truth reference for comparisons to the various estimation techniques. To bridge the gap between typical $R_\lambda$ of DNS and atmospheric flows, we use high-resolution measurements of the longitudinal velocity components of the Variable Density Turbulence Tunnel (VDTT) (Bodenschatz et al., 2014; Sinhuber, 2015; Küchler et al., 2019) at various Taylor Reynolds number $R_\lambda$ between 140 and 6000. The impact of turbulence intensity, large-scale random sweeping velocities, size of averaging window, Reynolds number and also possible experimental imperfections, such as anemometer misalignment are investigated in detail. Our work aims to be a step towards the goal of extracting the time-dependent energy dissipation rate from non-ideal naturally-occurring turbulent flows mitigating the impact of non-ideal features of the flow, e.g., anisotropy or inhomogeneity. In Sect. 2, we first define the central statistical quantities and the individual methods for estimating the energy dissipation rate in detail. An analysis of the individual methods including discrepancies, errors due to finite turbulence intensity, and alignment errors are discussed in Sect. 3 followed by a summary of our findings.

## 2 Methods

Let $\boldsymbol{u}(\boldsymbol{x},t) = u_1(\boldsymbol{x},t)\boldsymbol{e}_1 + u_2(\boldsymbol{x},t)\boldsymbol{e}_2 + u_3(\boldsymbol{x},t)\boldsymbol{e}_3$ denote the three-dimensional velocity vector of the turbulent flow, where $\boldsymbol{x} = x_1\boldsymbol{e}_1 + x_2\boldsymbol{e}_2 + x_3\boldsymbol{e}_3$ are the components of the Cartesian coordinate system and $t$ is the time. We assume that the streamwise direction of the global-mean flow $\boldsymbol{U}$ is in the direction of $\boldsymbol{e}_1$ such that $\boldsymbol{U} = U\boldsymbol{e}_1$ is (by definition) constant in space and time. We refer to $\boldsymbol{e}_1$ as the *longitudinal* direction and the components normal to that, i.e. $\boldsymbol{e}_2$ and $\boldsymbol{e}_3$, as the *transverse* directions of the flow. As mentioned earlier, many experimental setups record only a one-dimensional flow velocity at one location and as a function of time. We consider this one-dimensional velocity time record to be in the longitudinal flow direction unless otherwise stated, e.g. when the probe misalignment is investigated. In the following, we first introduce different averaging





principles that can be used to analyze turbulence statistics and Taylor's frozen hypothesis, and then present the commonly used methods for extracting the energy dissipation rate. A preliminary introduction of the basic statistical description of turbulent flows is provided in the appendix (Sec. A) for the sake of completeness.

## 2.1 On Averaging, Reynolds Decomposition and Taylor's Hypothesis

Most methods used to retrieve the dissipation rate require spatially resolved velocity statistics although the velocity is recorded only at a single point and as a function of time in many experiments. Therefore, prior to estimating the energy dissipation rate, the one-dimensional velocity time-record should be first mapped onto a spatially resolved velocity field. This is achieved by invoking Taylor's hypothesis, which requires a Reynolds decomposition of the velocity time-record by separating the velocity fluctuations from the mean velocity. To perform the Reynolds decomposition, we first have to clarify what is meant by the
mean velocity.

Generally, we have to distinguish between the global mean velocity $\boldsymbol{U} = \langle \boldsymbol{u}(\boldsymbol{x},t)\rangle = U\boldsymbol{e}_1$, the volume-averaged velocity $\langle \boldsymbol{u}(\boldsymbol{x},t)\rangle_R$ over a sphere of radius $R$, the time-averaged velocity $\langle \boldsymbol{u}(\boldsymbol{x},t)\rangle_\tau$ over a time interval $\tau$, and the ensemble-averaged velocity $\langle \boldsymbol{u}(\boldsymbol{x},t)\rangle_N$ over $N$ realizations (Wyngaard, 2010; Pope, 2000, among others). In this work, $\langle \cdot \rangle$ denotes the global mean, i.e. for infinitely large averaging windows in time or space. Thus, $\boldsymbol{U}$ is by definition independent of time and space, which in re-
ality is valid only when $\boldsymbol{u}(\boldsymbol{x},t)$ is statistically stationary and homogeneous. Implicitly, $\langle \boldsymbol{u}(\boldsymbol{x},t)\rangle_R = 3/(4\pi R^3) \iiint_0^R \mathrm{d}\boldsymbol{x}\, u(\boldsymbol{x},t)$ and $\langle \boldsymbol{u}(\boldsymbol{x},t)\rangle_\tau = \frac{1}{\tau}\int_{-\tau/2}^{\tau/2} \mathrm{d}t'\boldsymbol{u}(\boldsymbol{x},t')$ are, respectively, local volume and time averages as both $R$ and $\tau$ are typically finite. In the limit of $R,\tau \to \infty$, $\langle \boldsymbol{u}(\boldsymbol{x},t)\rangle_R$ and $\langle \boldsymbol{u}(\boldsymbol{x},t)\rangle_\tau$ tend to $\boldsymbol{U}$. For repeatable experiments where identical experimental conditions are guaranteed, $\langle \boldsymbol{u}(\boldsymbol{x},t)\rangle_N$ tends to $\boldsymbol{U}$ when $N \to \infty$.

The mean of a one-dimensional velocity time-record in the longitudinal direction $U_\tau$ here is defined by

$$U_\tau = \langle u_1(t)\rangle_\tau = \frac{1}{\tau}\int_{-\tau/2}^{\tau/2} \mathrm{d}t'u_1(t'), \tag{1}$$

such that the global mean $U = \lim_{\tau\to\infty} U_\tau$, where $\tau$ is the *averaging window*. Here, the velocity "time-record" means that the measurement is made at a single Eulerian point in the flow, so that the dependence on $\boldsymbol{x}$ in the longitudinal velocity component $u_1$ is eliminated. It should be noted that the global mean of the transverse velocity will be equal to zero, i.e. $\langle u_{2,3}(t)\rangle_\tau = 0$ when $\tau \to \infty$, since here it is assumed that they are orthogonal to the mean flow direction. According to the
Reynolds decomposition, the longitudinal velocity time record is composed of the mean velocity $U$ and the random velocity fluctuation component $u_1'(t) = u_1(t) - U$ so that the mean of the longitudinal velocity fluctuations $\langle u_1'(t)\rangle = 0$. This is also true for other components of the velocity.

In certain circumstances, it is possible to map $u_1'(t)$ from time to space coordinates by applying the Taylor's (frozen-eddy) hypothesis (Taylor, 1938; Wyngaard, 2010), which relates temporal and spatial velocity statistics. Taylor argues that eddies
can be regarded as *frozen* in time if they are passing the probing volume much faster than they evolve in time. This is the case if the turbulence intensity $I = \sigma_{u_1'}/U$ is much smaller than the unity, i.e. $I \ll 1$, where $\sigma_{u_1'} = \langle u_1'^2\rangle^{1/2}$ is the Root-Mean-





Square (RMS) velocity fluctuation. Then, the series of time lags $\Delta t = t - t_0$ relative to the start time $t_0$ is mapped onto a distance vector with $\boldsymbol{x} = \boldsymbol{x}_0 + U \Delta t \, \boldsymbol{e}_1$ (Taylor, 1938), where $\boldsymbol{x}_0$ is the initial position at time $t_0$. This approach is found to be reliable for $I \lesssim 0.25$ (Nobach and Tropea, 2012; Wilczek et al., 2014; Risius et al., 2015) while it has been shown to fail when

$I > 0.5$ (Willis and Deardorff, 1976). The application of Taylor's hypothesis is inaccurate in case of large-scale variations of the velocity fluctuation field comparable to the mean velocity, which are known as "random sweeping velocity" (Kraichnan, 1964; Tennekes, 1975) and which can be approximated by the turbulence intensity (Wilczek et al., 2014). Complicating the estimation of the mean velocity, random sweeping causes the mean energy dissipation rate to be consistently overestimated (Lumley, 1965; Wyngaard and Clifford, 1977).

One way to cope with non-stationary velocity time records is to evaluate the mean velocity for a subset of this signal. If the averaging time $\tau$ is finite, the time average $U_\tau$ may differ from the mean velocity $U$ causing a systematic bias in the subsequent data analysis. The estimation variance of the time average $U_\tau$ can be analytically expressed as (Wyngaard, 2010; Pope, 2000, among others)

$$\langle (U_\tau - U)^2 \rangle \approx \frac{2 \langle u_1'^2 \rangle T}{\tau}, \tag{2}$$

where $T$ is the integral time scale and $\langle u_1'^2 \rangle$ the variance of the velocity time series. Notably, the size of the averaging window has to be large enough such that it fulfills $\langle u_1'(t) \rangle_\tau \approx 0$ to apply the Reynolds decomposition. This expression can be converted to space invoking Taylor's hypothesis.

## 2.2   Estimating the Energy Dissipation Rate

The energy dissipation rate can be derived from various statistical quantities. A non-exhaustive list of the most common

methods applicable to single-point measurements is shown in Table 1. Details of selected methods considered in this study are presented in the following subsections. If not explicitly mentioned, the averages denoted with $\langle \cdot \rangle$ are defined globally.



| range | Dissipation estimate (Eq.) | symbol | definition | assumption |
|---|---|---|---|---|
| dissipative sub-range | instantaneous (3) | $\epsilon_0$ | $2\nu\left(s_{ij}s_{ij}\right)$ | |
| | (local) volume average (4) | $\langle\epsilon(\boldsymbol{x},t)\rangle_R$ | $\frac{3}{4\pi R^3}\iiint_{\mathcal{V}(R)}\epsilon_0(\boldsymbol{x}+\boldsymbol{r},t)\mathrm{d}\boldsymbol{r}$ | SHI |
| | (longitudinal) gradient (6) | $\epsilon_G$ | $15\nu\left\langle\left(\frac{\partial u_1'(\boldsymbol{x})}{\partial x_1}\right)^2\right\rangle$ | SHI |
| | 2nd-order SF (dissipation range) * | $\epsilon_{D2}$ | $15\nu D_{LL}(r)/r^2$ | SHI, $r \lesssim \eta_K$ |
| | zero-crossings * | $\epsilon_+$ | $15\pi^2\nu\langle u_1'^2\rangle N_L^2$ | SHI |
| inertial sub-range | 4/5 law * ((7), $n=3$) | $\epsilon_{I3}$ | $-5/4 D_{LLL}(r)/r$ | SHI, K41 |
| | 2nd-order SF (inertial range) ((7), $n=2$) | $\epsilon_{I2}$ | $\left(D_{LL}(r)/C_2\right)^{3/2}/r$ | SHI, K41 |
| | spectral (9) | $\epsilon_S$ | $\left(\frac{\kappa_1^{5/3}E_{11}(\kappa_1)}{18/55C_K}\right)^{3/2}$ | SHI, K41 |
| | cutoff filter * | $\epsilon_C$ | $\left(\frac{2}{3}\frac{2\langle u_C'^2\rangle}{18/55C_K\left(\kappa_{1,\text{low}}^{-2/3}-\kappa_{1,\text{up}}^{-2/3}\right)}\right)^{3/2}$ | SHI, K41 |
| energy injection scale | scaling argument (10) | $\epsilon_L$ | $C_\epsilon\sigma_{u_1'}^3/L_{11}$ | SHI |
| | global mean (5) | $\langle\epsilon\rangle$ | $\lim_{R\to\infty}\langle\epsilon_0(\boldsymbol{x},t)\rangle_R$ | SHI |

**Table 1.** Various definitions of the energy dissipation rate from the dissipative and inertial sub-range to the energy injection range. Here, the definitions for various dissipation estimates are given in the space or wavenumber domain where $\nu$ is the viscosity, $s_{ij}$ is the velocity fluctuation strain rate tensor, $R$ is the radius of the averaging volume $\mathcal{V}(R)$ (window size for 1D data), $u_1'(\boldsymbol{x})$ is the longitudinal velocity fluctuation field along $x_1$, $D_{L...L}(r)$ is the $n$th-order longitudinal structure function for distance $r$, $\langle u_1'^2\rangle$ is the variance of $u_1'(\boldsymbol{x})$, $\sigma_{u_1'}$ is the standard deviation of $u_1'(\boldsymbol{x})$, $N_L$ is the number of zero crossings of a velocity fluctuation signal per unit length, $C_2 \approx 2$, $E_{11}(\kappa_1)$ is the one-dimensional energy spectrum with wavenumber $\kappa_1$, $C_K \approx 1.5$, $\langle u_C'^2\rangle$ is the variance of a band-pass filtered signal for wavenumbers $\kappa_1 \in [\kappa_{1,\text{low}}, \kappa_{1,\text{up}}]$, $C_\epsilon$ is the dissipation constant, $L_{11}$ is the longitudinal integral scale, and $\eta_K$ is the Kolmogorov length scale. Dissipation estimates indicated with * are not considered in detail in this work. The assumptions of stationarity (S), homogeneity (H), local isotropy (I) and Kolmogorov's second similarity hypothesis from 1941 (K41) are represented by their individual abbreviations. References are given in the corresponding sections in the main text.

### 2.2.1 Dissipative sub-range

Proceeding from the Navier-Stokes equations for an incompressible, Newtonian fluid, the instantaneous energy dissipation rate is given by (e.g. Pope, 2000; Davidson, 2015)

$$\epsilon_0(\boldsymbol{x},t) = 2\nu\left(S_{ij}S_{ij}\right). \tag{3}$$

The contribution of the fluctuating part to the energy dissipation is much larger than the contribution of the mean flow in the case of high-Reynolds number turbulent flows (Pope, 2000; Elsner and Elsner, 1996). Hence, the instantaneous energy dissipation rate can be defined in terms of the velocity fluctuations only (Pope, 2000) where, in Eq. (3) and Table 1, $S_{ij}$ is replaced by the fluctuation strain rate tensor $s_{ij} = (\partial u_i'(\boldsymbol{x},t)/\partial x_j + \partial u_j'(\boldsymbol{x},t)/\partial x_i)/2$.

Averaged over a sphere with radius $R$ and volume $\mathcal{V}(R)$, the (local) volume average of the instantaneous energy dissipation rate is (Pope, 2000)

$$\epsilon_R(\boldsymbol{x},t) = \langle\epsilon_0(\boldsymbol{x},t)\rangle_R = \frac{3}{4\pi R^3}\iiint_{\mathcal{V}(R)}\epsilon_0(\boldsymbol{x}+\boldsymbol{r},t)\mathrm{d}\boldsymbol{r}. \tag{4}$$



The local volume average $\epsilon_R(\boldsymbol{x}, t)$ converges to the global mean energy dissipation rate if $R$ tends to infinity (Pope, 2000):

$$\langle \epsilon \rangle = \lim_{R \to \infty} \langle \epsilon_0(\boldsymbol{x}, t) \rangle_R = -\nu \lim_{|\boldsymbol{r}| \to 0} \partial^2_{r_j} R_{ii}(\boldsymbol{r}, t), \tag{5}$$

where the right-hand-side follows from partial integration. In experiments, it is often not possible to measure $\epsilon_0(\boldsymbol{x}, t)$. Under the assumption of statistically homogeneous and isotropic turbulence, the volume/time averaged energy dissipation rate are typically inferred from one-dimensional surrogates (Taylor, 1935; Elsner and Elsner, 1996; Siebert et al., 2006; Almalkie and de Bruyn Kops, 2012; Champagne, 1978; Donzis et al., 2008, among others), such as from the longitudinal velocity gradient (hence, the subscript G):

$$\epsilon_G = -15\nu \lim_{|\boldsymbol{r}| \to 0} \partial^2_{r_1} R_{11}(\boldsymbol{r}, t) = 15\nu \left\langle \left( \frac{\partial u'_1(\boldsymbol{x}, t)}{\partial x_1} \right)^2 \right\rangle = \frac{15\nu}{U^2} \left\langle \left( \frac{\partial u'_1(t)}{\partial t} \right)^2 \right\rangle, \tag{6}$$

where the mapping between space and time domains is possible by applying Taylor's hypothesis if $\sigma_{u'_1}/U \ll 1$ (Siebert et al., 2006; Muschinski et al., 2004). The relationship shown in Eq. (6) is often called the "direct" method in the literature (Muschinski et al., 2004; Siebert et al., 2006, e.g.) and requires a spatial resolution higher than the Kolmogorov length scale $\eta_K$ to be accurate to $\sim 10\%$ (cf. Fig. A8). The deviation of $\epsilon_G$ from its global mean $\langle \epsilon \rangle$ depends quadratically on the turbulence intensity

(Lumley, 1965; Wyngaard and Clifford, 1977; Champagne, 1978; Muschinski et al., 2004).

### 2.2.2 Inertial sub-range: indirect estimate of energy dissipation rate

Kolmogorov's second similarity hypothesis from 1941 (Kolmogorov, 1941) provides another method for estimating the energy dissipation rate in the inertial range. Based on the inertial range scaling of the $n$th-order longitudinal structure function, the mean energy dissipation rate can be calculated by (Pope, 2000)

$$D_{L \ldots L}(r) = C_n (\epsilon_{In} r)^{\zeta_n} \Leftrightarrow \epsilon_{In} = \left( \frac{D_{L \ldots L}(r)}{C_n} \right)^{1/\zeta_n} \frac{1}{r}, \tag{7}$$

where $C_n$ is a constant, e.g. $C_2 \approx 2$ (Pope, 2000), and $\zeta_n = n/3$ according to K41 by dimensional analysis. In practice, $\epsilon_{I2}$ (Table 1) is retrieved either by fitting a constant to the compensated longitudinal second-order structure function $D_{LL}(r)$, $n = 2$ in Eq. (7), or a power law ($\propto r^{2/3}$) to the inertial range of $D_{LL}$, defined in Eq. (A3), if the inertial range is pronounced over at least a decade. Accounting for intermittency, the scaling exponent of the $n$th-order structure function is modified to

$\zeta_n = \frac{n}{3}[1 - \frac{1}{6}\mu(n-3)]$ where $\mu$ is the internal intermittency exponent (Kolmogorov, 1962). The inertial range is bounded by the energy injection scale $L$ at large scales and by the dissipation range at small scales. That is why the fit-range has to be chosen such that $\eta_K \ll r \ll L$. If the inertial range is not sufficiently pronounced, the extended self similarity may be used to extend the inertial range (Benzi et al., 1993b, a). Otherwise, $\epsilon_{I2}$ can also be approximated by the maximum of Eq. (7) (for $n = 2$) within the same range as before. This is possible because the maximum lies on the plateau in case of a perfect K41

inertial range scaling.


In the inertial range, the transverse second-order structure function $D_{NN}(r)$ is equal to $4D_{LL}(r)/3$ in a coordinate system where $\boldsymbol{r} = r\boldsymbol{e}_1$ is parallel to the longitudinal flow direction (Pope, 2000) highlighting the importance of the measurement direction.

### 2.2.3 Inertial sub-range: spectral method

According to K41 (Kolmogorov, 1941), the inertial sub-range of the energy spectrum function scales as $E(\kappa) \propto \langle\epsilon\rangle^{2/3}\kappa^{-5/3}$ with the wavenumber $\kappa$ by dimensional analysis. In isotropic turbulence, the energy spectrum function can be converted into a one-dimensional energy spectrum $E_{11}(\kappa_1)$, see Eq. (A7). The wavenumber space is not directly accessible from one-dimensional velocity time-records. Relying on Taylor's hypothesis, the one-dimensional energy spectrum $E_{11}(\kappa_1)$ transforms to the frequency domain with $F_{11}(f) = 2\pi E_{11}(\kappa_1)/U$ where $\kappa_1 = 2\pi f/U$ (e.g. Wyngaard and Clifford, 1977; Oncley et al.,

1996) yielding:

$$F_{11}(f) = 18/55C_K\left(\frac{U}{2\pi}\epsilon_S\right)^{2/3}f^{-5/3}, \tag{8}$$

which yields

$$\epsilon_S = \frac{2\pi}{U}\left(\frac{f^{5/3}F_{11}(f)}{18/55C_K}\right)^{3/2}, \tag{9}$$

with the Kolmogorov constant $C_K = 1.5$ (Sreenivasan, 1995; Pope, 2000). Depending on the Fourier transform convention,

the prefactor of $C_K$, i.e. in the convention used here $18/55$ has to be adapted accordingly (Wyngaard, 2010, e.g.). Applying Taylor's hypothesis to a flow with a randomly-sweeping mean velocity causes the Kolmogorov constant to be systematically overestimated whereas the scaling of power-law spectra remains unaffected (Wilczek and Narita, 2012; Wilczek et al., 2014). Hence, Eq. (9) is still valid for a randomly-sweeping mean velocity although $\epsilon_S$ is overestimated if $C_K$ is not corrected for random sweeping.

$F_{11}$ has the units of a power spectral density $\mathrm{m^2\,s^{-1}}$ and $\langle u_1'^2\rangle = \int_0^\infty F_{11}(f)\mathrm{d}f$. Under the assumption of Kolmogorov scaling in the inertial sub-range, this identity can be adopted to estimate the mean energy dissipation rate from low and moderate resolution velocity measurements of a finite averaging window (Fairall et al., 1980; Siebert et al., 2006; O'Connor et al., 2010; Wacławczyk et al., 2017).

### 2.2.4 Energy injection scale

In equilibrium turbulence, the rate at which turbulent kinetic energy is transported across eddies of a given size is constant in the inertial range assuming high enough Reynolds numbers (e.g. Lumley, 1992). By dimensional argument, this rate is proportional to $u^3(l)/l$, where $u(l)$ is the characteristic velocity scale of eddies of length $l$. Considering the integral scale $L_{11}$ and its characteristic velocity scale $u(L_{11})$, namely the RMS velocity fluctuation $\sigma_{u_1'}$, the mean energy dissipation rate can be





calculated by (Taylor, 1935)

$$\epsilon_L = C_\epsilon \frac{\sigma_{u_1'}^3}{L_{11}}, \tag{10}$$

where $C_\epsilon$ is the dissipation constant and for time- and space-varying turbulence, it depends on both initial as well as boundary conditions and the large-scale structure of the flow (Sreenivasan, 1998; Sreenivasan et al., 1995; Burattini et al., 2005; Vassilicos, 2015). $C_\epsilon$ is found to be about 0.5 for shear turbulence (Sreenivasan, 1998; Pearson et al., 2002) and 1.0 (Sreenivasan, 1984; Sreenivasan et al., 1995) or 0.73 Sreenivasan (1998) for grid turbulence. In this work $C_\epsilon$ is assumed to be 0.5 which holds approximately in a variety of flows (Risius et al., 2015; Sreenivasan, 1995, and references therein).

Usually, the longitudinal integral length scale $L_{11}$ is defined as (Pope, 2000)

$$L_{11} = \lim_{r_0 \to \infty} \int_0^{r_0} \mathrm{d}r \, f(r) = \frac{\pi E_{11}(0)}{2\langle u_1'^2 \rangle}, \tag{11}$$

However, due to experimental limitations, $r_0$ is often given by the first zero-crossing of $f(r)$ in both laboratory and *in situ* measurements (Risius et al., 2015, e.g.), or, alternatively, by the position where $f(r) = 1/e$ (Tritton, 1977; Bewley et al., 2012). Griffin et al. (2019) carried out an integration for $r \to \infty$ performing an exponential fit in the vicinity of $f(r) = 1/e$. Notably, $E_{11}(0) = \int_0^\infty \mathrm{d}\kappa E(\kappa)/\kappa$ so that the estimation of $L_{11}$ from the power spectrum is only recommended if $E(\kappa) = \frac{1}{2}\kappa^3 \frac{\mathrm{d}}{\mathrm{d}\kappa}\left(\frac{1}{\kappa}\frac{\mathrm{d}E_{11}(\kappa)}{\mathrm{d}\kappa}\right)$ (Pope, 2000) is accurately determined like in DNS. This approach not only requires a fully resolved velocity measurement but also a well converged $E_{11}(\kappa_1)$ as the conversion is highly sensitive to statistical scatter. Ultimately, the choice of $L_{11}$ strongly affects $\epsilon_L$. In this work, we integrate $f(r)$ to the first zero-crossing because it does not depend on assumptions on the decay of $f(r)$ and the choice of the fit-range.

## 2.3 Simulations of homogeneous isotropic turbulence

In this study, the direct numerical simulations of statistically homogeneous isotropic turbulent flow with $74 \leq \mathrm{R}_\lambda \leq 321$ are used as the basis for evaluating the different methods for determining the dissipation rate, see Table 2. Thereby, the performance of the different methods to estimate the energy dissipation rate is not affected by violating fundamental assumptions, e.g., anisotropy or inhomogeneity. The simulations are carried out with the parallelized solver TurTLE (Lalescu et al., 2022), which solves the Navier-Stokes equations on a periodic domain using a pseudo-spectral method with a third-order Runge-Kutta time stepping. Here, we use a forcing scheme with a fixed energy injection rate on large scales. With this, we reach a statistically stationary state of homogeneous isotropic turbulence within approximately two to three integral times.

To mimic an ensemble of single-point measurements, we introduced 1000 virtual probes into the flow (one-way coupled, i.e. without back-reaction on the flow) , which move with a given constant speed in randomly-directed straight paths to record the local flow velocity. We assume that the virtual probe records idealized velocity time series neglecting the effect of transfer functions (e.g. Horst and Oncley, 2006; Freire et al., 2019, regarding sonic anemometry) or noise (Lenschow and Kristensen,



| case id | box size | $R_\lambda$ | $\dot{E}$ [c.u.] | $k_{\max}\eta_K$ | $\eta_K$ [c.u.] | $I$ | $L_{11}/\eta_K$ | $\mathcal{L}/L_{11}$ | $N_p$ [#] |
|---------|----------|-------------|------------------|------------------|-----------------|-----|-----------------|----------------------|-----------|
| DNS 1.1 | 512 | 74 | 0.4 | 3 | 0.015 | 0.01 | 41.2 | 161 | 10000 |
| DNS 1.2 | 512 | 74 | 0.4 | 3 | 0.015 | 0.05 | 41.4 | 160 | 10000 |
| DNS 1.3 | 512 | 74 | 0.4 | 3 | 0.015 | 0.10 | 41.3 | 160 | 10000 |
| DNS 1.4 | 512 | 74 | 0.4 | 3 | 0.015 | 0.24 | 41.3 | 21 | 10000 |
| DNS 1.5 | 512 | 74 | 0.4 | 3 | 0.015 | 0.50 | 41.4 | 16 | 10000 |
| DNS 2.0 | 1024 | 142 | 0.4 | 3 | 0.007 | 0.11 | 99.0 | 332.8 | 1000 |
| DNS 2.1 | 1024 | 219 | 0.4 | 3 | 0.007 | 0.01 | 147.8 | 15.6 | 1000 |
| DNS 2.2 | 1024 | 217 | 0.4 | 3 | 0.007 | 0.06 | 147.6 | 15.7 | 1000 |
| DNS 2.3 | 1024 | 216 | 0.4 | 3 | 0.007 | 0.11 | 147.9 | 15.6 | 1000 |
| DNS 2.4 | 1024 | 212 | 0.4 | 3 | 0.007 | 0.27 | 146.8 | 15.7 | 1000 |
| DNS 2.5 | 1024 | 207 | 0.4 | 3 | 0.007 | 0.53 | 145.5 | 15.8 | 1000 |
| DNS 3.1 | 2048 | 302 | 0.5 | 3 | 0.003 | 0.01 | 260.9 | 13.6 | 1000 |
| DNS 3.2 | 2048 | 299 | 0.5 | 3 | 0.003 | 0.05 | 258.2 | 13.8 | 1000 |
| DNS 3.3 | 2048 | 295 | 0.5 | 3 | 0.003 | 0.11 | 254.8 | 14.0 | 1000 |
| DNS 3.4 | 2048 | 314 | 0.5 | 3 | 0.004 | 0.26 | 275.6 | 20.2 | 1000 |
| DNS 3.5 | 2048 | 321 | 0.5 | 3 | 0.004 | 0.53 | 282.9 | 14.7 | 1000 |

**Table 2.** Parameter overview for each DNS. $R_\lambda$ is the Taylor scale Reynolds number, $\dot{E}$ the energy injection rate in code units (c.u.), $k_{\max}$ the largest resolved wavenumber, $\eta_K$ the Kolmogorov length scale, $I = \sigma_{u_1'}/U$ is the turbulence intensity, $L_{11}$ the longitudinal integral length scale derived from $E(\kappa)$, $\mathcal{L}$ is the average probe track distance and $N_p$ the number of virtual probes. The turbulence intensity $I$ is controlled by setting the probe mean velocity where $\sigma_{u_1'} \approx 1$ is the root mean square longitudinal velocity fluctuation.

1985; Antonia, 2003; Lewis et al., 2021). While the root mean square velocity fluctuation is determined by the Navier-Stokes simulation, we can control the mean flow speed through the speed of the virtual probe. The range of used constant speeds

corresponds to turbulence intensities of 1-50%. Along the trajectories, we then sample the local three-dimensional velocity field (see Fig. 1) as well as the velocity gradient field, where we use spline-interpolation of order 5 to determine values in between grid points, see also (Lalescu et al., 2010, 2022). By projecting the velocity vector on the direction of the trajectory, $e_1$, and the orthogonal directions, $e_2$ and $e_3$, we split the velocity field in longitudinal and transverse components, respectively. From the sampled velocity gradient tensor, we compute the local instantaneous dissipation $\epsilon_0$. The time step is limited either

by the stability requirements of the flow solver or, for smaller turbulence intensities, by the required sampling frequency to capture the underlying flow. Here, we choose the time step such that the distance traveled by the probe within one step is around a tenth of the grid spacing, $U\Delta t \approx 0.1\Delta x$. The grid spacing $\Delta x$ is chosen such that the highest wavenumber $k_{\max}$ satisfies $k_{\max}\eta_K \approx 3$.

Using Taylor's hypothesis, the longitudinal velocity time series correspond on average to $\sim 3000\eta_K$ (for more details see

Table 2) so that second- and third-order moments of both longitudinal velocity fluctuations and increments are reasonably converged (see Fig. A3). To estimate $\epsilon_{I3}$, $\epsilon_{I2}$ and $\epsilon_S$, the longitudinal structure functions are evaluated for scales $20\eta_K \leq r \leq 500\eta_K$ or in frequency domain for $\frac{U}{500\eta_K} \leq f \leq \frac{U}{20\eta_K}$. The ground-truth reference for the mean energy dissipation rate per virtual probe is given by $\langle\epsilon_0(\boldsymbol{x},t)\rangle_{\text{VP}}$, i.e. the average of the dissipation field along the trajectory of each virtual probe. The global mean energy dissipation rate can be approximated by the ensemble average of all $\langle\epsilon_0(\boldsymbol{x},t)\rangle_{\text{VP}}$ from all virtual probes,

i.e. $\langle\langle\epsilon_0(\boldsymbol{x},t)\rangle_{\text{VP}}\rangle_N$.





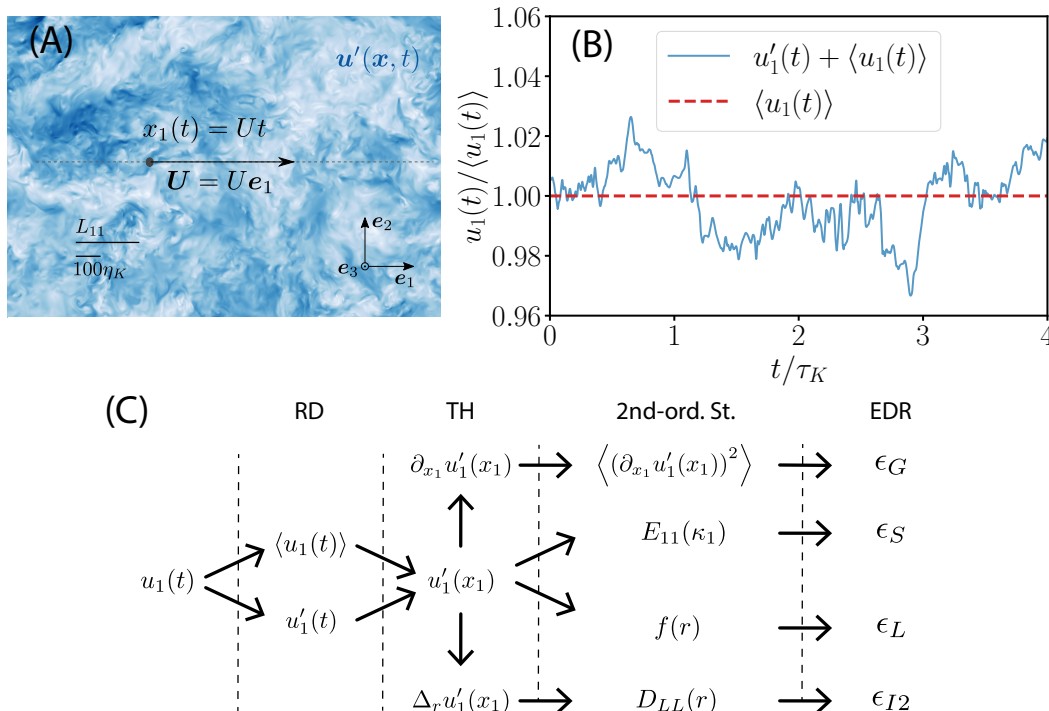

**Figure 1.** From single-point velocity time records to the energy dissipation rate. (A) Virtual probe sampling the three-dimensional velocity field of the DNS 3.1 (see Table 2) in time and space at a mean velocity $U$ along its $e_1$-direction corresponding to turbulence intensity of 1%. (B) *One*-dimensional velocity time series $u_1(t)$ (blue solid) with corresponding time average $U = \langle u_1(t) \rangle$ (Eq. (1), red dashed line) of the same DNS 3.1, where both $u_1(t)$ and $\langle u_1(t) \rangle$ are re-scaled by $U$. (C) Visualization of the workflow from one-dimensional velocity time record $u_1(t)$ to the energy dissipation rate via different methods. First, $u_1(t)$ is decomposed in its mean and fluctuating part according to Reynolds decomposition (RD). Then, the velocity time series is converted into a one-dimensional velocity field invoking Taylor's hypothesis (TH). Subsequently, second-order statistics (2nd-ord. St.) of the longitudinal velocity fluctuations, its increments and first spatial derivative are inferred from which the energy dissipation rate is estimated with the help of different methods.

## 2.4 Variable Density Turbulence Tunnel (VDTT)

To evaluate the performance of different methods at Reynolds numbers applicable to atmospheric flows, we use the high-resolution hot-wire measurements of the longitudinal velocity components in the MPIDS VDTT (VDDT, Bodenschatz et al., 2014). The VDTT datasets used here are associated with $R_\lambda$ between 147 and 5864, which enables us to bridge the gap between DNS ($74 \leq R_\lambda \leq 321$) and atmospheric $R_\lambda \sim \mathcal{O}(10^3)$.

The VDTT is a recirculating wind tunnel where the working gas $SF_6$ is pressurized up to 15 bar. The VDTT has a horizontal length of 11.68 m and an inner diameter of 1.52 m where the rotation frequency of the fan sets the mean flow velocity ranging from 0.5 m/s to 5.5 m/s (Bodenschatz et al., 2014). Long-range correlations of the turbulent flow determine its anisotropy. These long-range correlations are shaped with the help of an active grid consisting of 111 independently rotating winglets (Küchler et al., 2019; Küchler, 2021). The angular orientation of each winglet $\vartheta(t, y, z)$ can be individually adjusted in space





and time by a control software depending on spatio-temporal grid-correlations (Griffin et al., 2019):

$$\vartheta(t,y,z) = \int K(t',y',z')A(t-t',y-y',z-z')\mathrm{d}t'\mathrm{d}y'\mathrm{d}z', \tag{12}$$

where $A$ is a three-dimensional random matrix of winglet angles and $K$ is the correlation kernel. The turbulence in the experiments with $R_\lambda \in \{4141, 5006, 5865\}$ was driven in an anisotropic way (Küchler, 2021).

Longitudinal velocity fluctuations are temporally recorded with $30\,\mu$m to $60\,\mu$m long nanoscale thermal anemometry probes (NSTAP; Bailey et al., 2010; Vallikivi et al., 2011, among others) or a $450\,\mu$m long conventional hot-wire from Dantec (Jørgensen, 2001) corresponding to a resolution of $< 3\eta_K$ and $< 5\eta_K$, respectively (Küchler et al., 2019) at variable distances from the active grid ranging from $\approx 6-9\,$m. The velocity measurements have been extensively characterized in terms of the mean flow profiles (Küchler, 2021) as well as the decay of turbulent kinetic energy (Sinhuber et al., 2015; Sinhuber, 275   2015) exposing velocity probability distribution functions (PDF) being flatter than Gaussian (Küchler, 2021). The inertial range scaling exponent $\zeta_2$ of the longitudinal second-order structure function is in agreement with Kolmogorov's revised phenomenology from 1962 ($\zeta_2 = 0.693 \pm 0.003$ for $R_\lambda > 2000$) for a large variety of wake generation schemes (Küchler et al., 2020). In the case of hot-wire measurements in the VDTT, the ground-truth energy dissipation rate for a given averaging window $R$ is given by the gradient method $\langle \epsilon_G \rangle_R$, which converges fastest as shown in Sec. 3.5.

## 280  2.5  Strategies for the evaluation of systematic and random errors

Virtual probes record one-dimensional time records of DNS longitudinal velocity, from which the mean energy dissipation rate can be estimated by various methods and compared with the energy dissipation rate obtained directly from the DNS dissipation field. Generally, there are two different errors when estimating the mean energy dissipation rate, namely the systematic errors and random errors. The latter is related to the estimation variance of the mean energy dissipation rate, i.e. the statistical scatter 285   of the $\langle \epsilon \rangle$-estimates around the true mean. The systematic error of the mean energy dissipation rate expresses itself in a non-vanishing ensemble average of the deviations from the ground-truth.

Systematic errors are an inherent feature of the methods used for estimating the dissipation rate, but are also affected by experimental limitations and imperfections such as averaging windows and finite turbulence intensity parameterized by $R$ and $I$, respectively. One way to estimate these errors is to compare the estimated mean energy dissipation rate for a given 290   averaging window $R$ to the ground-truth of the DNS defined by the mean energy dissipation rate per virtual-probe track, i.e., $\langle \epsilon_0(\boldsymbol{x},t) \rangle_{\mathrm{VP},R}$. Another possibility would be to compare the estimates to the ensemble average of the mean energy dissipation rate from all virtual probes, i.e., $\langle \langle \epsilon_0(\boldsymbol{x},t) \rangle_{\mathrm{VP},R} \rangle_N$, where $N = 1000$ is the total number of virtual probes. Either of these possibilities is valid and would be interesting to understand. However, our analysis shows that the second approach is associated with a slightly higher absolute value and a slightly higher standard deviation. For that reason, we have chosen the second 295   approach to make a conservative assessment of the systematic errors, i.e., we compare the estimates of each method against





$\langle\langle\epsilon_0(\boldsymbol{x},t)\rangle_{\mathrm{VP},R}\rangle_N$, by

$$\beta_i = \frac{\langle\epsilon_i\rangle_R}{\langle\langle\epsilon_0(\boldsymbol{x},t)\rangle_{\mathrm{VP},R}\rangle_N} - 1\,, \tag{13}$$

where $i \in \{G, I3, I2, S, L\}$ and $\langle\epsilon_i\rangle_R$ is the estimate of the energy dissipation rate via method $i$ under the experimental limitation and imperfection such as size of averaging window or finite turbulence intensity. To distinguish between the different
error terms in this manuscript, we refer to $\beta$ as "reference-compared" systematic error.

In addition, the systematic error can be evaluated by comparing the estimates of the energy dissipation rate obtained by a method with imperfect data against the estimates obtained by the same method with optimal data. We denote these types of errors with $\delta$ and refer to them as "self-compared" errors. An experimental imperfection we considered here is the sensor misalignment, which is a non-zero angle of incidence $\theta$ between the longitudinal flow direction that sensor expect and $U$. To
investigate the isolated effect of sensor misalignment, we consider a specific set of DNS with constant turbulence intensity ($I = 1\%$) and the entire track length for each virtual probe. The self-compared systematic error of each method due to misalignment is defined as

$$\delta_i(\theta) = \frac{\epsilon_i(\theta)}{\epsilon_i(0)} - 1\,, \tag{14}$$

where $\epsilon_i(\theta)$ is the estimate of the energy dissipation rate via method $i \in \{G, I3, I2, S, L\}$ from data with misalignment $\theta$ and
$\epsilon_i(0)$ is the estimated dissipation rate from the same method and flow conditions but with an aligned sensor, i.e. $\theta = 0$.

Estimates of the mean energy dissipation rate are susceptible not only to systematic errors, but also to random errors due to statistical uncertainty. For the averaging window, errors given by Eq. (13) would be the best indicator of systematic errors. However, random errors due to size of averaging window can also be significant. When the spatial averaging window $R$ (or temporal averaging window $\tau$) is finite, we capture the self-compared random error for each individual method by

$$\delta_i(R) = \sqrt{\left\langle\left(\frac{\langle\epsilon_i\rangle_R}{\langle\langle\epsilon_i\rangle_R\rangle_N} - 1\right)^2\right\rangle_N}\,, \tag{15}$$

where $\langle\epsilon_i\rangle_R$ is the local mean energy dissipation rate based on the averaging window $R$ normalised by its ensemble average, i.e. $\langle\langle\epsilon_i\rangle_R\rangle_N$. Eq. (15) indeed calculates the standard deviation of the normalized $\langle\epsilon_i\rangle_R$, which is used here as a proxy for the random error. Table 3 provides an overview of the different error types and terminologies used here.

## 3  Results and Discussion

In the following, we first focus on the DNS data to calculate $\epsilon_G, \epsilon_{I3}, \epsilon_{I2}, \epsilon_S$, and $\epsilon_L$ from the entire longitudinal velocity time records of all virtual probes and compare these estimates against the ground-truth reference. Then, we systematically investigate the impact of turbulence intensity, (virtual) probe orientation, and averaging window size for all methods of interest. The



| Symbol | Definition | Equation |
|---|---|---|
| $\beta_i$ | reference-compared systematic error, i.e. relative to ground-truth reference $\langle\langle\epsilon_0(\boldsymbol{x},t)\rangle_R\rangle_N$ | (13) |
| $\delta_i(\theta)$ | self-compared (systematic) error of each method at a given misalignment angle $\theta$ relative to the estimates provided by the same method but at $\theta = 0$ | (14) |
| $\delta_i(R)$ | self-compared (random) error at a given averaging window of $R$ or $\tau$ relative to the average value from all virtual probes at the same averaging window of $R$ or $\tau$ | (15) |

**Table 3.** Overview of investigated errors and their definitions. $i \in \{G, I3, I2, S, L\}$, where $G$ stands for gradient method, $I3$ for 4/5 law, $I2$ for second-order structure function in the inertial range, $S$ for the spectral method, and $L$ for the scaling argument. The averaging window is denoted spatially by $R$ and temporally by $\tau$. The misalignment angle is represented by $\theta$.

influence of flow Reynolds number on the presented results are then discussed by taking into account the VDTT data together with the DNS data. Finally, we provide a proof of concept for a time-dependent dissipation rate calculation by comparing the

dissipation time series measured by $\epsilon_G$, $\epsilon_{I2}$, and $\epsilon_L$ and its coarse-grained surrogate. In the following, we use the definitions of systematic and random errors as mentioned in Sec. 2.5 and Table 3.

### 3.1 Verification of the analytical methods and a first insight into their performance under ideal conditions

To verify the implementation of our methods, only data from cases with a low turbulence intensity of 0.01 and an averaging window covering the entire size of the probe track are used in this section. Furthermore, $\epsilon_{I2}$ and $\epsilon_{I3}$ are obtained by a fit

according to Eq. (7) with $n = 2$ and $n = 3$, respectively, in the inertial range with $r \in [20\eta_K, 500\eta_K]$ for DNS 2.1 and 3.1. Analogously, $\epsilon_S$ is inferred from the inertial range fit, Eq. (9), in the range $f \in [U/(500\eta_K), U/(20\eta_K)]$. For DNS 1.1 with $R_\lambda = 74$, due to the absence of an inertial range for low Taylor-scale Reynolds number (see Fig. A7), the maximum of Eq. (7) is used to infer $\epsilon_{I2}$ and $\epsilon_{I3}$ instead of fitting the inertial range.

The distribution of the mean energy dissipation rate estimated by $\epsilon_G, \epsilon_{I2}, \epsilon_{I3}, \epsilon_S$, and $\epsilon_L$ for each probe at $R_\lambda = 302$ is

shown in Fig. 2. Estimations for other $R_\lambda$ are shown in supplementary Fig. A1. The ground-truth reference for the mean energy dissipation rate for each probe is given by $\epsilon_{\text{ref}} = \langle\epsilon_0(\boldsymbol{x},t)\rangle_{\text{VP}}$ along the probe trajectory $\boldsymbol{x}$, which is the average of the instantaneous energy dissipation rate along the trajectory of each individual virtual probe (mean, median, standard deviation and range of $\beta_{\text{ref}}$: $0\%$, $-0.7\%$, $18.6\%$, $-50\% \ldots 68.2\%$ where $\beta_{\text{ref}} = \langle\epsilon_0(\boldsymbol{x},t)\rangle_{\text{VP}}/\langle\langle\epsilon_0(\boldsymbol{x},t)\rangle_{\text{VP}}\rangle_N - 1$). The best performing method is the gradient method $\epsilon_G$ (mean, median, standard deviation and range of $\beta_G$: $-0.5\%$, $1.7\%$, $19.3\%$, $-48.1\% \ldots 75.4\%$). The

range of $\beta_G$ is also very close to the range of $\beta_{\text{ref}}$. The method with highest error is $\epsilon_{I3}$ (mean, median, standard deviation, range of $\beta_{I3}$: $49.2\%$, $10.1\%$, $59.6\%$, $-93.1\% \ldots 822.2\%$). The superior performance of $\epsilon_G$ compared to others is mainly due to the fact that it relies on second-order dissipative statistics that can be captured with fast statistical convergence within a short sampling interval. Hence, the distribution of $\epsilon_G$ and $\epsilon_{\text{ref}}$ are similar. $\epsilon_{I3}$, on the other hand, relies on third-order moments of the velocity increments of inertial scales associated with slower statistical convergence compared to $\epsilon_G$. Therefore, $\epsilon_{I3}$ requires

longer velocity records than $\epsilon_G$ to converge under stationary conditions. For this reason, the third-order structure function is not considered further in this study, as one of the main objectives of this study is to evaluate different methods suitable for extracting the time-dependent energy dissipation rate.



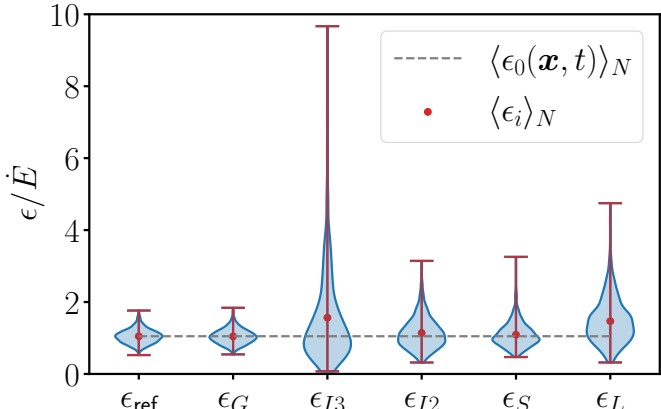

**Figure 2.** Validation of estimating the energy dissipation rate from $\epsilon_G, \epsilon_{I2}, \epsilon_{I3}, \epsilon_S$, and $\epsilon_L$ re-scaled by the energy injection rate $\dot{E}$. The data are taken from DNS 3.1 with 1000 probes, $R_\lambda = 302$, $I = 1\%$, $\theta = 0°$ and maximal available averaging window ($R \approx 3550\eta_K$). The ensemble mean of each method $\langle \epsilon_i \rangle_N$ is denoted by red dots where the whiskers extend from the minimal to maximal estimate of $\epsilon_i$ where $i \in \{G, I3, I2, S, L\}$. The reference mean energy dissipation rate for each probe is given by $\epsilon_{ref}$. The dashed line represents the re-scaled global mean energy dissipation rate of DNS 3.1 which is approximated by the ensemble average of the true mean energy dissipation rate along the trajectory of each virtual probe.

Fig. 2 also shows that the estimates of the energy dissipation rate provided by $D_{LL}(r)$ and $E_{11}(\kappa_1)$ are close to each other, which can be explained by the fact that they are both second-order quantities (in real and Fourier space, respectively) connected by $f(r)$. Furthermore, $\epsilon_{I2}$ tends to overestimate the energy dissipation rate as the mean is 8.8% higher than $\langle \epsilon(\boldsymbol{x},t) \rangle_N$ (median $\beta_{I2} \sim 3.1\%$, standard deviation 41.7%, $-69.6\% < \beta_{I2} < 199.8\%$). $\epsilon_S$ exhibits a similar overestimation (mean $\beta_S \sim 4.1\%$, median $\beta_S \sim -2.0\%$, standard deviation 32.1%, $-55.2\% < \beta_S < 210.4\%$), though to a lesser extent. However, $\epsilon_S$ depends much stronger on properly setting the fit-range than $\epsilon_{I2}$ (supplementary Fig. A2). The spectral method $\epsilon_S$ can differ by a factor of 2 from $\epsilon_{I2}$ depending on the high-frequency limit. This factor of 2 is in accordance with a comparison of $\epsilon_{I2}$ and $\epsilon_S$ by a linear fit resulting in a slope close to 0.5 (Akinlabi et al., 2019). In the DNS, the power spectrum is subject to strong statistical uncertainty at high frequencies without ensemble-averaging the spectra of each virtual probe or longer DNS runtimes. As the high-frequency limit of the inertial range of the spectrum is hardly distinguishable from its dissipation range, the choice of the fit-range range for $\epsilon_S$ is related to the fit-range of the longitudinal second-order structure function by $f \in [U/(500\eta_K), U/(20\eta_K)]$ as mentioned above. Wacławczyk et al. (2020) found that the estimation of the energy dissipation rate from the power spectral density is generally robust at small wavenumbers whereas the second-order structure function performs better at larger wavenumbers. With our choice of the fit-range $r \in [20\eta_K, 500\eta_K]$ for DNS 3.1 dataset shown in Fig. 2, we confirm that $\epsilon_{I2}$ is already reliable at the lower end of the inertial range where dissipative effects are negligible.

At last, $\epsilon_L$ overestimates $\langle\langle \epsilon_0(\boldsymbol{x},t) \rangle_{VP} \rangle_N$ by 40% on average (median $\beta_L \sim 31.5\%$, $-69.5\% < \beta_L < 352.9\%$). This systematic overestimation might be due to the difficulty in determining $L_{11}$ as different methods for estimating the integral length $L_{11}$ can contribute to the systematic bias of $\epsilon_L$. As mentioned above, we infer the longitudinal integral length from fitting $f(r)$ to the first zero crossing which yields, at least in the DNS of this work, a systematic underestimation, as illustrated in





Fig. A3. Figure A3 and A4 suggest that the scatter of $\epsilon_L$ is affected by the scatter of both $\sigma_{u'_1}$ and $L_{11}$. However, the accuracy of the dissipation constant $C_\epsilon$, which is a function of large-scale forcing and initial conditions (Vassilicos, 2015; Sreenivasan, 1998; Sreenivasan et al., 1995; Burattini et al., 2005), can potentially cause larger mean deviations of $\epsilon_L$ from $\langle\langle\epsilon_0(\boldsymbol{x},t)\rangle_{\text{VP}}\rangle_N$.

Advantageously, the large-scale estimate $\epsilon_L$ is applicable to low-resolution measurements and only weakly biased with respect to the ground-truth of DNS 1.1 where the variance is better converged (see Table 4 and Fig. A1). Table 4 gives an overview of the systematic errors of the different methods at different Reynolds numbers, showing that the above conclusions are also valid for lower $R_\lambda$.

| DNS | $\langle\beta_G\rangle_N$ | $\langle\beta_{I3}\rangle_N$ | $\langle\beta_{I2}\rangle_N$ | $\langle\beta_S\rangle_N$ | $\langle\beta_L\rangle_N$ |
|---|---|---|---|---|---|
| 1.1 | $-0.003 \pm 0.001$ | $0.132 \pm 0.005$ | $-0.047 \pm 0.002$ | $0.011 \pm 0.002$ | $-0.044 \pm 0.003$ |
| 2.1 | $-0.002 \pm 0.006$ | $0.506 \pm 0.038$ | $-0.011 \pm 0.014$ | $0.074 \pm 0.010$ | $0.313 \pm 0.017$ |
| 3.1 | $-0.005 \pm 0.006$ | $0.492 \pm 0.039$ | $0.088 \pm 0.013$ | $0.041 \pm 0.010$ | $0.400 \pm 0.020$ |

**Table 4.** The systematic error of each method $\beta_i$ relative to the global mean energy dissipation rate, Eq. (13), of each DNS where $i \in \{G, I3, I2, S, L\}$. The error is given by the standard error, which is defined as the standard deviation divided by the square root of the number of samples. In both DNS 2.1 and 3.1, $\epsilon_{I2}$ and $\epsilon_{I3}$ were obtained by fitting Eq. (7) for $n = 2$ and $n = 3$, respectively, in the range $r \in [20\eta_K, 500\eta_K]$. This fit-range is also used for calculating $\epsilon_S$ and it was converted into frequency domain by $f = U/r$, where $U$ is the mean velocity. In the case of DNS 1.1, the maximum of Eq. (7) was used to infer $\epsilon_{I2}$ due to the absence of a pronounced inertial range. We used the maximum available window size $R$ in all cases, fixed turbulence intensity $I = 1\%$ and considered perfect alignment, i.e. $\theta = 0°$.

## 3.2 Validity of Taylor's hypothesis and impact of random sweeping effects

A finite turbulence intensity causes the time to space conversion of velocity statistics to be inaccurate and, hence, affects the energy dissipation rate. In the case of high turbulence intensity ($I \gtrsim 0.5$), the eddy turnover time is on the same order as the advection time. As a result, the local speed and direction of the flow is significantly varying in time and space, which hinders the applicability of the Taylor's hypothesis. Here, we quantify the impact of random sweeping on the accuracy of determining the mean energy dissipation rate. Therefore, we set the mean speed of the virtual probes in each DNS so that the turbulence

intensity, and in consequence the random sweeping, is a control parameter.

Fig. 3 shows the systematic errors $\beta_i$ for $\epsilon_G$, $\epsilon_{I2}$, $\epsilon_S$, and $\epsilon_L$ at different turbulence intensities for DNS 3.1-5. For each virtual probe taken into account in Fig. 3, we used the entire time series so that the size of the averaging window is maximal. While each method has a different systematic error and scatter, Fig. 3 indicates that the mean relative deviation of each estimate from the global mean $\langle\langle\epsilon_0(\boldsymbol{x},t)\rangle_{\text{VP}}\rangle_N$ increases with turbulence intensity. This is particularly strong for the gradient method. For

$I = 1\%$ and $I = 10\%$, the gradient method has the lowest scatter in terms of the standard deviation $\sigma_{\beta_G}$ (19.3% and 27.3%) and the lowest systematic error in terms of the $\langle\beta_G\rangle_N$ ($-0.5\%$ and $6.1\%$), respectively. At higher turbulence intensities, $\epsilon_{I2}$ is the least affected method with $\langle\beta_{I2}\rangle_N = 6.5\%$ and $\sigma_{\beta_{I2}} = 37.2\%$ for $I = 25\%$ as well as $\langle\beta_{I2}\rangle_N = 24.5\%$ and $\sigma_{\beta_{I2}} = 56.9\%$ for $I = 50\%$. At the highest turbulence intensities, both $\epsilon_L$ and $\epsilon_S$ are associated with lower mean $\beta$ than that of $\epsilon_G$.

The fraction of track samples that can lead to a deviation of larger than 100% increases from 0% to $\sim 60\%$ for $\epsilon_G$ as the

turbulence intensity increases from 1% to 50%. We hypothesize that these deviations of the mean are the result of random sweeping effects, which limit the applicability of Taylor's hypothesis. In frequency space, Taylor's hypothesis establishes a one-to-one mapping between the frequency and the streamwise wavenumber, i.e. $\omega = \kappa_1 U$. As the turbulence intensity grows,



a randomly sweeping mean velocity smears out this correspondence between frequencies and wavenumbers. For the spectrum, this smearing out effectively moves energy from larger scales to smaller and less energetic ones. Therefore, it leads to an

overestimation in the inertial and dissipation range of the spectrum, thus, affecting the inertial range and gradient method. To visualize this overestimation, we evaluate the effect of random sweeping on the spectrum (Eq. (B2)) numerically for different turbulence intensities at the example of a model spectrum (Eq. (B4)). The result is shown in Fig. 4, where the spectrum is pre-multiplied by $\kappa_1^2$ to later highlight the effect on the gradient method. Here, the overestimation is most pronounced in the dissipative range.

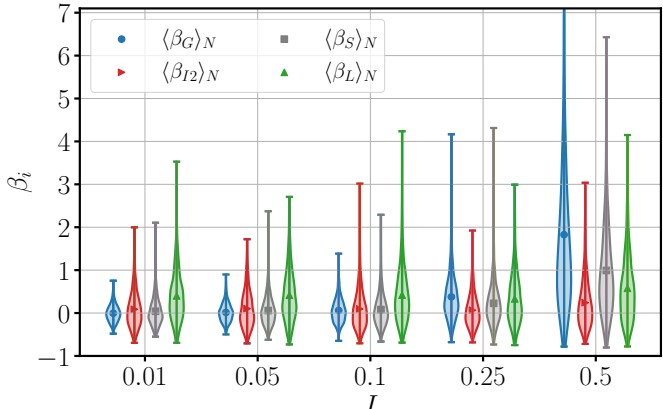

**Figure 3.** Systematic error $\beta_i$, Eq. (13), as a function of turbulence intensity $I \in \{0.01, 0.05, 0.1, 0.25, 0.5\}$ for $\epsilon_G$ (●), $\epsilon_{I2}$ (►), $\epsilon_S$ (■), and $\epsilon_L$ (▲). The energy dissipation rates are estimated from each longitudinal velocity time series of DNS 3.1-5 with ideal alignment ($\theta = 0°$) where the maximal available window size was used. The fit-range for the inertial range of the power spectral density is chosen to be within $U/(500\eta_K) \leq f \leq U/(20\eta_K)$ where $\eta_K$ is the Kolmogorov length scale, and, equivalently in space domain, $20\eta_K \leq r \leq 500\eta_K$ for the longitudinal second-order structure function. The upper limit of the y-axis is chosen to be 7.1 for improving the plot visibility (there are some outliers of $\epsilon_G$ for $I = 50\%$).

To quantify the impact of random sweeping on estimates of $\epsilon$, we first consider the influence of random sweeping on the gradient method. For the gradient method, Lumley (1965) and Wyngaard and Clifford (1977) have shown that in isotropic turbulence random sweeping leads to an relative deviation of the volume-averaged mean energy dissipation rate by

$$\frac{\epsilon_G}{\langle\epsilon\rangle} - 1 = \beta_G = 5I^2 \quad \text{with} \quad \epsilon_G = 15\nu \int \kappa_1^2 E_{11}(\kappa_1) d\kappa_1 = \langle\epsilon\rangle[1 + 5I^2]. \tag{16}$$

We illustrate this result in the appendix Sec. B, where we consider a model wavenumber-frequency spectrum (Wilczek and

Narita, 2012; Wilczek et al., 2014), which is based on the same modeling assumptions used in Wyngaard and Clifford (1977). Due to the $\kappa_1^2$-weighting of the gradient method, the mean dissipation rate estimate is highly sensitive to the viscous cutoff of the energy spectrum, which is overestimated by random sweeping effects, see Fig. 4. As a consequence, deviations of the estimated dissipation rate are growing rapidly with turbulence intensity. In the right panel of Fig. 4, we compare the effect of random sweeping on the gradient method obtained with a model spectrum, the one computed by Lumley (1965), and the

observed deviations by measurements of the virtual probes in a DNS flow; here shown are the DNS 3.1-5. In fact, the estimate





from (Lumley, 1965) can explain the magnitude of deviations observed by the virtual probes in case of $\epsilon_G$ up to $I = 25\%$. The strong deviation of $\beta_G$ at $I = 50\%$ is likely due to the sensitivity of the gradients on the space-to-time conversion via Taylor's hypothesis: at high turbulence intensities, the mean velocity becomes smaller compared to the fluctuations. Therefore, the error of estimating the mean velocity due to finite averaging window, Eq. (2), increases relatively to the mean velocity.

Larger relative errors in the estimated mean velocity lead - applying Taylor's hypothesis - to both under- and overestimated spatial gradients for the individual averaging windows, additional to the effect of random sweeping. Similarly this results in an additional overestimation of the dissipation rate. These deviations do not appear in evaluating random sweeping effects based on a model spectrum as there the mean velocity is a parameter we choose.

Now let us consider the two inertial sub-range methods. Here, as one can see in Fig. 3 and 4, the increase of the mean

relative deviation, $\beta_i$, is less pronounced. In the inertial sub-range, random sweeping causes an overestimation of the spectrum of merely several percent while the inertial range scaling is preserved as shown in Wyngaard and Clifford (1977); Wilczek et al. (2014). As both the second-order structure function and the spectral method are based on the inertial sub-range of the energy spectrum, the effect of a randomly sweeping mean velocity is expected to be small on $\epsilon_{I2}$ and $\epsilon_S$. Here, the overestimation of the spectrum can be used to express the relative systematic deviation of both $\epsilon_{I2}$ and $\epsilon_S$ for different turbulence intensities

analytically:

$$\frac{\epsilon_{I2/S}}{\langle \epsilon \rangle} - 1 = \beta_{I2/S} = (C_T(I))^{3/2} - 1 \quad \text{with} \quad C_T(I) = \frac{5}{6} \int\limits_0^\infty \mathrm{d}y \left[ \mathrm{erf}\left(\frac{y+1}{\sqrt{2}I}\right) - \mathrm{erf}\left(\frac{y-1}{\sqrt{2}I}\right) \right] y^{2/3}, \tag{17}$$

where $C_T(I)$ quantifies the spectral overestimation as function of mean wind and fluctuations defined as in (Wilczek et al., 2014). In Fig. 4B we compare the observed deviations from the DNS to Eq. (17). This shows that Eq. (17) underestimates $\beta_{I2}$ for $I \in \{0.01, 0.05, 0.1\}$ (i.e. DNS 3.1, 3.2 and 3.3). The underestimation is most likely due to additional random errors

associated with finite averaging window lengths. It is obvious from Table 2 that DNS 3.3 has statistically the shortest probe tracks $\sim 3440\eta_K$ (DNS 3.1: $\sim 3550\eta_K$, DNS 3.2: $\sim 3560\eta_K$). Nonetheless, $\beta_{I2}$ matches the prediction of Eq. (17) for $I \in \{0.25, 0.5\}$ where the corresponding probe tracks statistically amount to $\sim 5570\eta_K$ and $\sim 4260\eta_K$, respectively. The effect of the averaging window size on $\epsilon_{I2}$ is explored in Sec. 3.4. We conclude that Eq. (17) can be used to estimate the error introduced by random sweeping of $\epsilon_{I2}$.

For the spectral method, Eq. (17) underestimates the relative error $\beta_S$ for all turbulence intensities. This may be due to the strong dependence of $\epsilon_S$ on the $U$-based fitting range, i.e., $f \in [U/(500\eta_K), U/(20\eta_K)]$, which can differ significantly between virtual probes at high turbulence intensities. Further work is needed to assess the dependence of the spectral method on the choice of the fit-range for finite turbulence intensities.

Overall, random sweeping effects explain why the gradient method is more sensitive to turbulence intensity than inertial-

range methods. Here, random sweeping accurately captures the deviations of the second-order structure function method as a function of turbulence intensity whereas it can only partially account for the observed deviations for the spectral method.





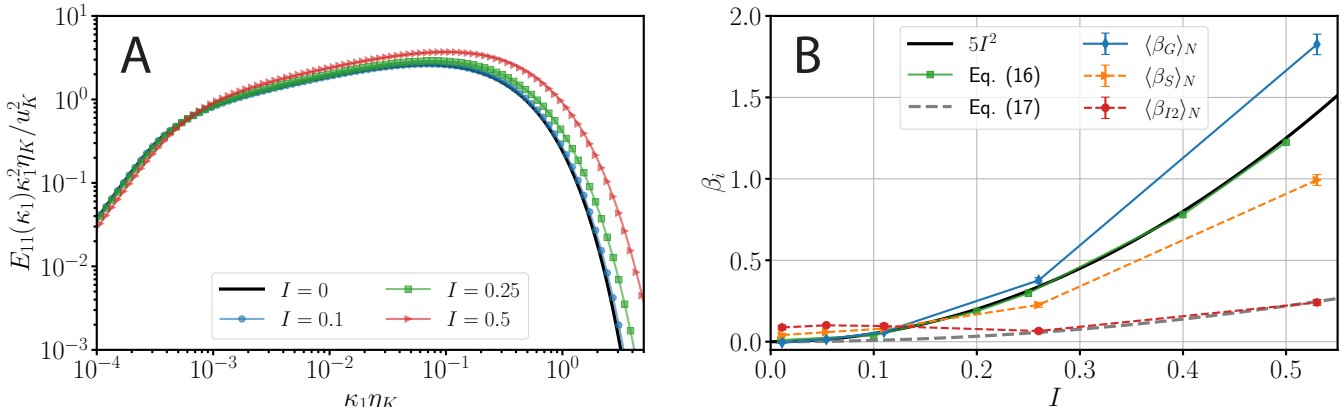

**Figure 4.** The effects of random sweeping on the energy dissipation, Eq. (B2), assuming a model spectrum. (A) Premultiplied energy spectrum with random sweeping effects for turbulence intensities $I \in \{0.1, 0.25, 0.5\}$ where the original energy spectrum corresponds to $I = 0$. In Eq. (B4), $\langle \epsilon \rangle$ is chosen to be equal to 1, the Kolmogorov length scale to $\eta_K \approx 0.00038$ m. $u_K = (\nu \langle \epsilon \rangle)^{1/2}$ is the Kolmogorov velocity scale. (B) Systematic over-prediction illustrated by the relative error $\beta_i$, Eq. (13), at different turbulence intensities. The systematic over-prediction by (Lumley, 1965) (solid black) matches with the numerically obtained systematic error $\beta_G$ for the gradient method relative to the ground-truth reference $\langle \epsilon \rangle$ by using the model spectrum (Eq. (B3), green squares). Both reasonably estimate the data obtained from DNS 3.1-5 (blue diamonds) up to a turbulence intensity of $I = 25\%$. Also, we show the systematic over-prediction of inertial sub-range methods ($\beta_S$: orange triangles, and $\beta_{I2}$: red circles, both from Eq. (13)) compared to the analytically derived error obtained by the random sweeping model ($\beta_{I2,S}$, Eq. (17), grey dashed).

### 3.3 Probe Misalignment

In this section, we assess the influence of probe misalignment with respect to the mean flow direction on estimating the energy dissipation rate at the energy injection scale, the inertial range and the dissipation range. Here, we assume the angle $\theta$ between

the (virtual) anemometer and the global mean wind direction $\frac{U}{|U|}$ to be constant throughout the sampling trajectory. As it can be seen from Eq. (C5), $\epsilon_L$ depends on $\theta$. Then, the analytically derived error for $\epsilon_L$ due to misalignment of the sensor and the longitudinal wind direction is given by

$$\delta_L(\theta) = \frac{\epsilon_L(\theta)}{\epsilon_L(0)} - 1 = \frac{2}{\cos\theta(1 + \cos^2\theta)} - 1, \tag{18}$$

where $\epsilon_L(\theta)$ represents the energy dissipation that is derived given an angle of incidence $\theta$ and $\epsilon_L(0)$ is the reference value for

perfect alignment of the mean flow direction and the probe, i.e. when $\theta = 0$.

Analogously, the second-order structure function tensor is effected by misalignment (cf. appendix C). Thus, it can be shown that the analytically derived error $\delta_{I2}(\theta)$ as a function of $\theta$ reads

$$\delta_{I2}(\theta) = \frac{\epsilon_{I2}(\theta)}{\epsilon_{I2}(0)} - 1 = \left(\frac{4 - \cos^2\theta}{3}\right)^{3/2} \frac{1}{\cos\theta} - 1, \tag{19}$$



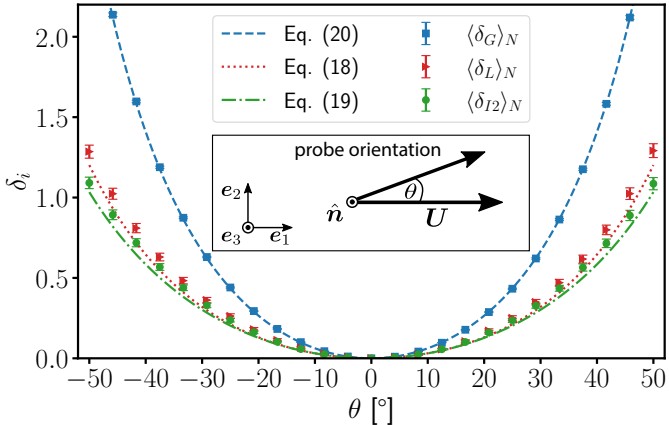

**Figure 5.** Influence of misalignment between probe orientation and the mean flow direction $U$ in terms of the average error of the energy dissipation rate $\delta_i(\theta)$ as a function of the angle of attack $\theta$. The energy dissipation rates are derived from DNS 3.1 with a turbulence intensity of 1%, $R_\lambda = 293$ and maximally available window size. The error bars are given by the standard error of the mean. The analytically derived errors $\delta_G(\theta)$, $\delta_L(\theta)$, and $\delta_L(\theta)$ are given by Eqs. (20), (18), and (19), respectively. The ordinate is limited from 0 to 2.2 to guarantee a better visibility for $\delta_L(\theta)$ and $\delta_{I2}(\theta)$. The inset visualizes the misalignment angle $\theta$ between the probe orientation and the mean flow direction $U$. The rotation axis is denoted by $\hat{n}$. As mentioned above, the mean flow direction $U$ is considered as the longitudinal direction of the flow.

where $\epsilon_{I2}(\theta)$ represents the energy dissipation that is derived given an angle of incidence $\theta$ and $\epsilon_{I2}(0)$ is the reference value

for perfect alignment of the mean flow direction and the probe. As outlined in Appendix C and with Eq. (5), the analytically

derived error of $\epsilon_G$ as a function of $\theta$ can be calculated to

$$\delta_G(\theta) = \frac{\epsilon_G(\theta)}{\epsilon_G(0)} - 1 = 2\left(\frac{1}{\cos^2\theta} - 1\right), \tag{20}$$

where $\epsilon_G(\theta)$ represents the energy dissipation that is derived given an angle of incidence $\theta$ and $\epsilon_G(0)$ is the reference value for

perfect alignment of the mean flow direction and the probe.

To compare the analytical expressions to DNS results, the sensing orientation of the virtual probes is rotated around the

$e_3$-axis in the coordinate system of each the virtual probe by an angle $\theta$ relative to their direction of motion, i.e. the $e_1$-axis.

Then, $\epsilon_L(\theta)$, $\epsilon_{I2}(\theta)$, and $\epsilon_G(\theta)$ are inferred from the new longitudinal velocity component. The ensemble-averaged relative

errors of the estimated energy dissipation rates $\delta(\theta)$ due to misalignment is shown as a function of $\theta$ in Fig. 5 in the range of

$\pm 50°$ both for DNS and the analytically derived Eqs. (20), (18), and (19). In general, the ensemble-averaged systematic errors

follow the analytically derived errors reliably in terms of the limits of accuracy for all $R_\lambda$ at turbulence intensity $I = 1\%$. The

longitudinal second-order structure function is the best performing method with a systematic error $\langle\delta_{I2}\rangle_N$ of lower than 20%

for $\theta \in [-25°, 25°]$, which increase to 100% at $\theta = \pm 50°$. $\langle\delta_L\rangle_N$ is similarly affected by misalignment but slightly larger than

$\langle\delta_{I2}\rangle_N$. Despite its rapid statistical convergence, $\epsilon_G$ is the most vulnerable method by misalignment compared to the other two

methods.

In experiments where the sensor can be aligned to the mean wind direction within $\theta \in [-10°, 10°]$ over the entire record

time, $\delta_i(\theta)$ is expected to be small. Further work is needed to evaluate the impact of a time-dependent misalignment angle





$\theta(t)$. We suppose that keeping the angle of attack $\theta$ fixed over the entire averaging window, here the entire time record of each probe, potentially leads to overestimation of $\delta_i(\theta)$ with $\theta$ being a function of time in practice.

### 3.4 Systematic errors due to finite averaging window size $R$

Here, our goal is to investigate how the accuracy of estimating the global mean energy dissipation rate depends on the averaging window size by investigating the associated systematic and random errors individually. To do this, we select an averaging window of size $R$ from the beginning of each track of virtual probes for case DNS 3.1. In this way, we obtain 1 subrecord for each virtual probe, which amounts to a total of 1000 subrecords for each averaging window $R$. From each of these subrecords a mean value of $\epsilon_0$ (i.e. $\langle\epsilon_0(\boldsymbol{x},t)\rangle_{\mathrm{VP},R}$), $\langle\epsilon_G\rangle_R$, $\langle\epsilon_L\rangle_R$ and $\langle\epsilon_{I2}\rangle_R$ are then evaluated. The smallest $R$ considered for these

analyses is $501\eta_K$, which is limited by the upper bound of the fitting range $r \in [20\eta_K, 500\eta_K]$ for estimating $\epsilon_{I2}$. The largest window size considered in this section is $3000\eta_K$, which is limited by the total length of the virtual-probe track (Table 2).

Before comparing estimates of the energy dispersion rate using different methods, let us first compare the locally averaged energy dispersion rate $\langle\epsilon_0(\boldsymbol{x},t)\rangle_{\mathrm{VP},R}$ with the instantaneous energy dispersion rate, which is shown in Fig. 6A. All averaging window sizes create PDFs with similar shape, but significantly different from the shape of the instantaneous field. The larger

the volume over which the dissipation field is averaged, the more the PDF($\langle\epsilon_0(\boldsymbol{x},t)\rangle_{\mathrm{VP},R}$) converges to a peak at the global mean energy dissipation rate normalised by $\dot{E}$, i.e. $\langle\epsilon_0(\boldsymbol{x},t)\rangle/\dot{E} \approx 1.0$.

We can further explore the influence of averaging window $R$ for each method by examining the distribution of systematic errors, i.e., $\beta_i$, as shown in Fig. 6B-D. First main point to note is the fact that all methods at small $R$ tend to peak at a dissipation rate lower than the global. Hence, the mean energy dissipation rate is most likely underestimated. All PDF($\beta_i(R)$) become

narrower and the mean relative errors $\beta_i(R)$ converge to 0 as $R$ increases. The second main point to consider is the statistical uncertainty, causing a random error in estimating the local mean energy dissipation rate $\langle\epsilon_0(\boldsymbol{x},t)\rangle_{\mathrm{VP},R}$. As it can be seen in Fig. 6B-D, the width of the distribution is wide with asymmetric long tails, especially for $\beta_{I2}$ and $\beta_L$. This is an indication that high random errors are to be expected in the estimation of the mean energy distribution rate.

### 3.5 Random errors due to finite averaging window size $R$

We now focus on random errors associated with $\epsilon_G$, $\epsilon_L$ and $\epsilon_{I2}$ analytically. We denote $\langle\epsilon_G\rangle_R$, $\langle\epsilon_L\rangle_R$ and $\langle\epsilon_{I2}\rangle_R$ the energy dissipation rates that are estimated for a longitudinal velocity time record for a window of size $R$. For the calculation of random errors caused by the choice of the size of the averaging window, we consider DNS 1.3, 2.3, and 3.3, as well as wind tunnel experiments that all have a comparable turbulence intensity of $I \approx 10\%$.

Both the second-order structure function, Eq. (A3), and the scaling argument, Eq. (10), depend on the variance $\langle u_1'^2\rangle$ of the

longitudinal velocity time record. $\epsilon_G$ is also related to $\langle u_1'^2\rangle$ through Eqs. (6) and (A1). The variance $\langle u_1'^2\rangle$ itself is subject to both systematic and random errors in case of a finite averaging window $R < \infty$. Assuming an ergodic, hence, a stationary velocity-fluctuation time-record with a vanishing mean, the systematic error in estimating the variance over an averaging

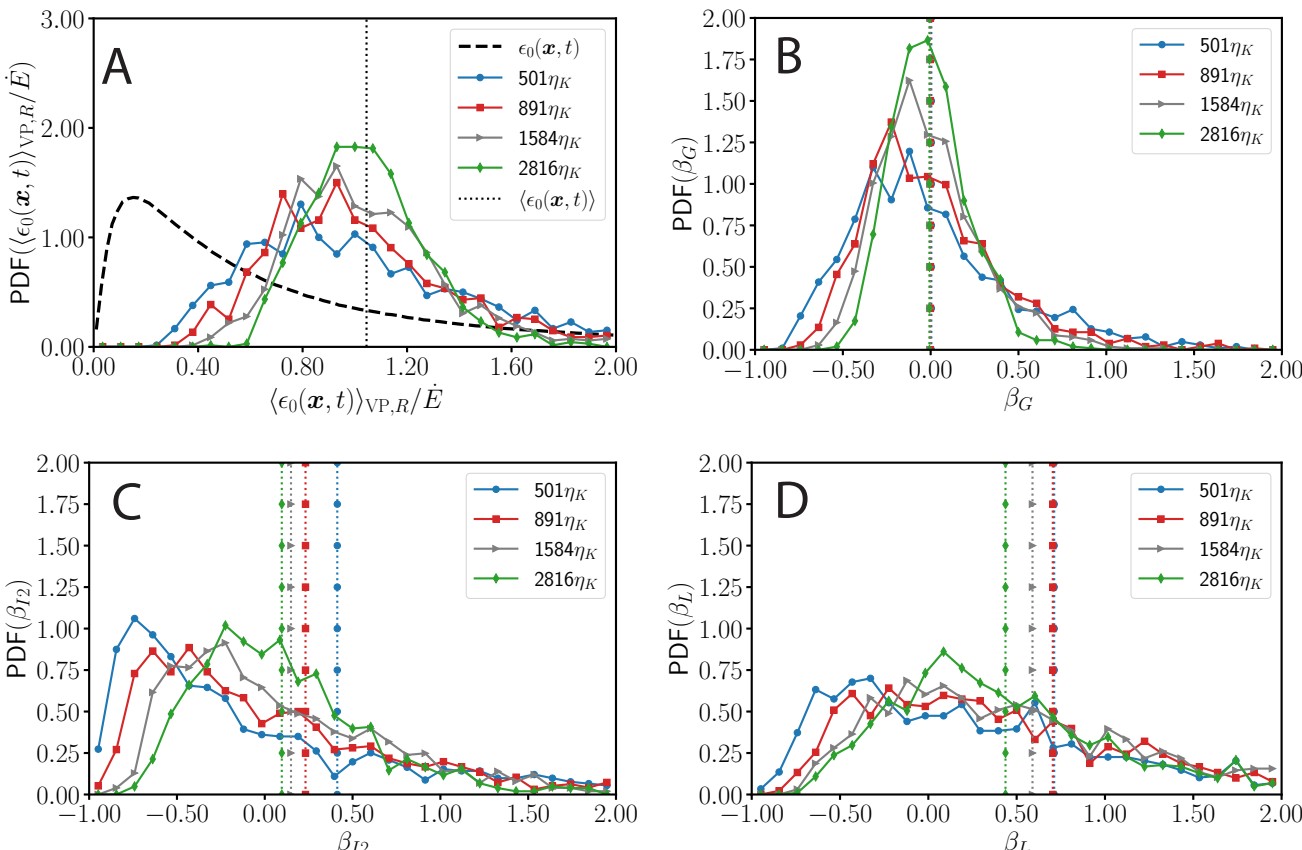

**Figure 6.** The effect of the averaging window size $R$ (A) on the distribution of $\langle \epsilon_0(\boldsymbol{x}, t)\rangle_{\mathrm{VP},R}/\dot{E}$; and on the accuracy of estimates obtained via (B) $\langle \epsilon_G \rangle_R$, (C) $\langle \epsilon_{I2} \rangle_R$, and (D) $\langle \epsilon_L \rangle_R$ in terms of the systematic errors $\beta_G$, $\beta_{I2}$, and $\beta_L$, respectively, from the ground-truth reference $\langle \epsilon_0(\boldsymbol{x}, t)\rangle_{\mathrm{VP},R}$ as given by Eq. (13). The velocity time records of the longitudinal component are taken from DNS 3.1 ($R_\lambda = 302$, $I = 1\%$, $\theta = 0°$). In (A), the distribution of the instantaneous dissipation rate $\epsilon_0(\boldsymbol{x}, t)/\dot{E}$ sampled by all virtual probes is shown by the dashed line, and the global average energy dissipation rate normalized by $\dot{E}$ is shown by the dotted vertical line. The other PDFs in A are from the local average of the energy dissipation rate obtained from a window of size $R$ at the beginning of each virtual probe, i.e., 1000 averaged values for a given $R$. In (B), (C) and (D) the vertical dotted lines correspond to ensemble averages of the systematic errors $\beta_i$. The ensemble average of $\beta_G$ slightly decreases from 0.4% for $R = 501\eta_K$ to $-0.7\%$ for $R = 2815\eta_K$ where the standard deviation of $\beta_G$ decreases from 50% to 22%. The ensemble average of $\beta_{I2}$ decreases from 41% to 10% and the standard deviation from 185% to 5%. $\beta_L$ exhibits stronger deviations (mean $\beta_L$ of $\sim 44\%$ and standard deviation $\sim 67\%$ for $R = 2816\eta_K$).



window of size $R$ is given by (following Lenschow et al., 1994, while applying Taylor's hypothesis)

$$\Delta_{\langle u_1'^2 \rangle} = \left\langle \frac{\langle u_1'^2 \rangle_R}{\langle u_1'^2 \rangle} - 1 \right\rangle_N \approx -2\frac{L_{11}}{R}, \tag{21}$$

where $\langle u_1'^2 \rangle_R$ is the estimated variance based on the (finite) averaging window $R$, $\langle u_1'^2 \rangle$ is the true variance and it is assumed $R \gg L_{11}$. The always negative error predicted by equation (21) indicates that, for finite averaging window sizes, the variance $\langle u_1'^2 \rangle$ is always statistically underestimated, which agrees with Figure A3A. Eq. (21) furthermore indicates that the systematic error of the variance estimates can be neglected for *sufficiently* long averaging windows $R \gg L_{11}$.

The variance estimates are also subject to statistical uncertainty, which is also known as the random error of variance
estimation (Lenschow et al., 1994). Assuming that $u_1'(t)$, which has a zero mean, can be modeled by a stationary Gaussian process and that its autocorrelation function is sufficiently well represented by an exponential, the random error of estimating the variance can be expressed as (following Lenschow et al., 1994, while applying Taylor's hypothesis)

$$e_{\mathrm{rand}} = \sqrt{\left\langle \left( \frac{\langle u_1'^2 \rangle_R - \langle \langle u_1'^2 \rangle_R \rangle_N}{\langle u_1'^2 \rangle} \right)^2 \right\rangle_N} \approx \sqrt{\left\langle \left( \frac{\langle u_1'^2 \rangle_R}{\langle \langle u_1'^2 \rangle_R \rangle_N} - 1 \right)^2 \right\rangle_N} \approx \sqrt{\frac{2L_{11}}{R}}, \tag{22}$$

where it is assumed $R \gg L_{11}$ such that the systematic error can be neglected and, hence, $\langle \langle u_1'^2 \rangle_R \rangle_N \approx \langle u_1'^2 \rangle$. Here, $\langle \langle u_1'^2 \rangle_R \rangle_N$
is the ensemble average of the variance estimates $\langle u_1'^2 \rangle_R$ for an averaging window $R$. It can be seen that $e_{\mathrm{rand}}$ is larger than the systematic error, (21), when $R > L_{11}$.

Consequently, the estimation of the mean energy dissipation rate by the scaling argument, Eq. (10), is affected by the (absolute) random error of the variance estimation given by the product of $e_{\mathrm{rand}}$ and $\left\langle \langle u_1'^2 \rangle_R \right\rangle_N$. Invoking the Gaussian error propagation, the analytically derived error reads

$$\delta_L(R) = \frac{1}{\langle \epsilon_L \rangle_R} \frac{\partial \langle \epsilon_L \rangle_R}{\partial \langle u_1'^2 \rangle_R} \underbrace{\underbrace{e_{\mathrm{rand}}}_{\text{rel. rand. err. of } \langle u_1'^2 \rangle_R} \left\langle \langle u_1'^2 \rangle_R \right\rangle_N}_{\text{abs. rand. err. of } \langle u_1'^2 \rangle_R} = \frac{3}{2}\sqrt{\frac{2L_{11}}{R}}. \tag{23}$$

$\delta_L(R)$ is a relative error, hence the prefactor $1/\langle \epsilon_L \rangle_R$. Notably, $\delta_L(R)$ scales as $R^{-1/2}$.

Similarly, the longitudinal second-order structure function is also affected by the estimation variance of the variance,

$$e_{D_{LL}} = \sqrt{\left\langle \left( \frac{D_{LL}(r; R)}{\langle D_{LL}(r; R) \rangle_N} - 1 \right)^2 \right\rangle_N} = \sqrt{\left\langle \left( \frac{2\langle u_1'^2 \rangle_R (1 - f(r))}{2\langle \langle u_1'^2 \rangle_R \rangle_N (1 - f(r))} - 1 \right)^2 \right\rangle_N} \approx \sqrt{\frac{2L_{11}}{R}}, \tag{24}$$

where $D_{LL}(r; R)$ is the longitudinal second-order structure function evaluated over an averaging window of size $R$ and under
the assumption that the longitudinal auto-correlation function $f(r)$ is well converged over the range of the averaging window.





Thus, the uncertainty of estimating the variance propagates to $\langle \epsilon_{I2} \rangle_R$ relying on $D_{LL}(r; R)$ (Eq. (7) for $n = 2$). The random error $\delta_{I2}(R)$ can be analytically inferred from the random error of the second-order structure function $\sigma_{D_{LL}}$ by Gaussian error propagation yielding

$$\delta_{I2}(R) = \frac{1}{\langle \epsilon_{I2} \rangle_R} \frac{\partial \langle \epsilon_{I2} \rangle_R}{\partial D_{LL}} e_{D_{LL}} \langle D_{LL}(r; R) \rangle_N = \frac{3}{2} \sqrt{\frac{2L_{11}}{R}}, \tag{25}$$

which shows that $\delta_{I2}(R)$ scales as $R^{-1/2}$ similar to $\delta_L(R)$. Considering Eqs. (6) and (A1), the gradient method can also be expressed as a function of the variance $\langle u_1^{'2} \rangle$. Hence, Gaussian error propagation yields:

$$\delta_G(R) = \frac{1}{\langle \epsilon_G \rangle_R} \frac{\partial \langle \epsilon_G \rangle_R}{\partial \langle u_1^{'2} \rangle_R} e_{\text{rand}} \left\langle \langle u_1^{'2} \rangle_R \right\rangle_N = -15\nu \frac{1}{\langle \epsilon_G \rangle_R} \lim_{|\boldsymbol{r}| \to 0} \partial_r^2 f(r) e_{\text{rand}} \left\langle \langle u_1^{'2} \rangle_R \right\rangle_N = \sqrt{\frac{2L_{11}}{R}}, \tag{26}$$

assuming $R \gg L_{11}$ such that the systematic error is negligible such that $\left\langle \langle u_1^{'2} \rangle_R \right\rangle_N \approx \langle u_1^{'2} \rangle$.

Equations (23), (25), and (26) are expressed as a function of $R$ and $L_{11}$, which do not reveal the dependency of random errors on the Reynolds number. In addition, this expression relies on large scales that depend on the scale of the energy injection, which makes it difficult to fairly compare the errors between different flows as it is not a universal feature. Therefore, we want to link the averaging window to the Kolmogorov length scale $\eta_K$, which only depends on the viscosity and the mean energy dissipation rate. We can rewrite these equations in terms of $\eta_K$, $R$ and $R_\lambda$ as follows:

$$\delta_{I2}(R) = \delta_L(R) = \frac{3}{2} \sqrt{\frac{2L_{11}}{R}} = \frac{3}{2} \sqrt{2 \frac{\eta_K}{R} \frac{L_{11}}{L} \left( \frac{3}{20} R_\lambda^2 \right)^{3/4}} \approx \frac{3}{2} \sqrt{\frac{\eta_K}{R} \left( \frac{3}{20} R_\lambda^2 \right)^{3/4}} \tag{27}$$

$$\delta_G(R) = \sqrt{\frac{2L_{11}}{R}} \approx \sqrt{\frac{\eta_K}{R} \left( \frac{3}{20} R_\lambda^2 \right)^{3/4}} \tag{28}$$

where we have invoked $L_{11}/L \sim 1/2$, which is valid at sufficiently high $R_\lambda$ and used the relationship $L/\eta_K = \left( \frac{3}{20} R_\lambda^2 \right)^{3/4}$ (Pope, 2000). Following the intuition, the longer the averaging window, the smaller the random error of each method.

Furthermore, Eqs. (27) and (28) provide a mean to choose a suitable averaging window size to achieve a given random error threshold $a$. Let $R_a$ be the averaging window of size $R$ such that $\delta_i(R) < a$. Then, the required averaging window $R_a$ for $\epsilon_{I2}$ and $\epsilon_L$ is

$$R_a/\eta_K = \frac{9}{4} \left( \frac{3}{20} R_\lambda^2 \right)^{3/4} \frac{1}{a^2}, \tag{29}$$

where the required averaging window size $R_a$ scales with $R_\lambda^{3/2}$. Similarly, the required averaging window for $\epsilon_G$ is

$$R_a/\eta_K = \left( \frac{3}{20} R_\lambda^2 \right)^{3/4} \frac{1}{a^2}. \tag{30}$$



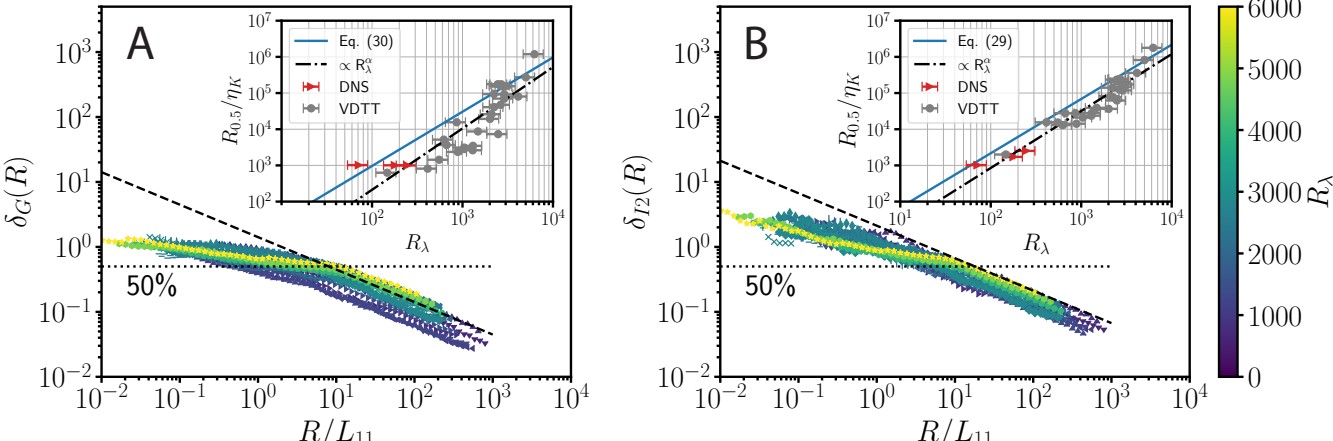

**Figure 7.** Random errors $\delta_G(R)$ (A) and $\delta_{I2}(R)$ (B) as a function of re-scaled averaging window size $R/L_{11}$ obtained from VDTT data at various $R_\lambda$ shown by the colorbar. The analytical results for $\delta_G(R)$ (A, Eq. (26)) and $\delta_{I2}(R)$ (B, Eq. (25)) are shown by the dashed black lines. The dotted black line annotated with "50%" in each subplot corresponds to 50% error-threshold. The insets show the sizes of the averaging windows in terms of $\eta_K$ when $\delta_{G,I2}(R) \leq 0.5$ as a function of Taylor microscale Reynolds number $R_\lambda$. The inset plots include data from both DNS (red triangles) and the VDTT (grey circles). DNS data used for the inset plots are from cases 1.3, 2.3 and 3.3 with $I = 10\%$ and $\theta = 0°$. The solid, blue lines show the prediction of the required averaging window according to Eq. (30) (A-inset) and Eq. (29) (B-inset). The black dash-dotted line in inset plots is a fit to the data: $\log R/\eta_K = \frac{3}{4}\log\frac{3}{20} - 2\log a_{\text{fit}} + \alpha\log R_\lambda$ yielding $\alpha = 1.70 \pm 0.18$ and $a_{\text{fit}} = 1.67 \pm 0.64$ (A-inset); $\log R/\eta_K = \log\frac{9}{4}\frac{3}{20}^{3/4} - 2\log a_{\text{fit}} + \alpha\log R_\lambda$ yielding $\alpha = 1.57 \pm 0.09$ and $a_{\text{fit}} = 0.95 \pm 0.32$ (B-inset).

For example, for the random errors of $\epsilon_{I2}$ and $\epsilon_L$ to be less than 10% at $R_\lambda = 1000$, the averaging window should be

$R \sim 2 \times 10^6 \eta_K \sim 2 \times 10^4 L_{11}$, while for $\epsilon_G$ the required averaging window is $R \sim 8 \times 10^5 \eta_K \sim 10^4 L_{11}$.

Figure 7 shows the empirical random errors $\delta_G(R)$ (Fig. 7A) and $\delta_{I2}(R)$ (Fig. 7B) as a function of the averaging window size for various $R_\lambda$ based on VDTT data (for $\epsilon_L$ see supplementary Fig. A5). To do this, we select an averaging window of size $R$, where $1000\eta_K < R < \mathcal{O}(10^6\eta_K)$, from the beginning of each $30\,\text{s}$ time-segment from the VDTT longitudinal velocities are recorded (a total of 47 to 597 time-segments depending on $R_\lambda$).

The scaling of $\delta_G(R)$ and $\delta_{I2}(R)$ is well predicted for $R \gtrsim 10L_{11}$ as expected from Eqs. (26) and (25) and the assumptions we made to derive them. However, for smaller $R$ a statistical convergence of $\epsilon_G$, $\epsilon_{I2}$ or $\epsilon_L$ against the mean energy dissipation rate cannot be expected, in particular when $R/L_{11} < 1$.

Furthermore, it is evident from Fig. 7 that the random errors do not fully collapse on each other for different Reynolds numbers and at a given $R/L_{11}$. Moving horizontally on a line of constant random error, e.g., the dashed line of 50% error, the

required window size increases with $R_\lambda$, as shown in the insets of Figs. 7A and B. Predictions of Eqs. (29) and (30) are also shown in these plots via solid/blue lines.

For both $\epsilon_G$ and $\epsilon_{I2}$, the theoretical expectation for $R_a$ tends to overestimate the actual averaging window size at which a random error of 50% is achieved. This overestimation is expected as the theoretical expectation for $R_a$, Eqs. (29) and (30), are derived assuming that large-scale quantities such as $f(r)$ and $L_{11}$ are fully converged. However, $\epsilon_G$ is technically relying on

small scales. $\epsilon_G$ depends on velocity fluctuation gradients, which are numerically obtained by central differences. Hence, each





increment in the velocity record contributes to the average in the gradient method, Eq. (6). In the case of $\epsilon_{I2}$, the number of possible increments reduces for larger separations for a finite averaging window. By definition, the exact computation of $L_{11}$ requires even a fully converged $f(r)$ for all $r$.

However, VDTT experiments with $R_\lambda > 3000$ underestimate the prediction of Eq. (21) by about a factor of 2. This is

particularly clear for $\epsilon_L$ shown in Fig. A5. This deviation at high $R_\lambda$ can be explained, at least in part, by the strong assumptions made for the derivation of the random errors, i.e., the equations (25), (23), and (26). In particular, for experiments with high Re in VDTT, the assumption of Gaussian velocity fluctuations with zero skewness is questionable, as shown in Fig. A6. Lenschow et al. (1994) has already established that the size of the averaging window for a skewed Gaussian process (see Eq. (19) in Lenschow et al., 1994) must be twice as large as for a Gaussian process with vanishing skewness. However, further work is

needed to investigate these deviations and improve the theoretical prediction.

### 3.6 Estimating the transient energy dissipation rate

As it has been shown in previous Figs. 6 and 7, both systematic and random error decrease with the size of the averaging window. For a correct estimate of the magnitude, it is therefore advantageous to choose the averaging window as large as possible, but this has the price that the transient trend smaller than the selected window size cannot be reproduced. In addition,

it is also important to know to what extent the estimated trend correlates with the actual trend. Given a certain averaging window size $R$, here, we empirically evaluate if trends in the coarse-grained time-series are physical or rather statistical. In other words, we ask the question if local estimates of the mean energy dissipation rate follow the ground-truth reference $\langle \epsilon_0(\boldsymbol{x}, t) \rangle_{\mathrm{VP},R}$ or not. Respecting the intermittent nature of turbulence and energy dissipation, the standard deviation of $\langle \epsilon_0(\boldsymbol{x}, t) \rangle_{\mathrm{VP},R}$ is a first proxy for the variability of the trend in $\langle \epsilon_0(\boldsymbol{x}, t) \rangle_{\mathrm{VP},R}$. Hence, detecting the true trend requires that $\beta_i$ and $\delta_i(R)$ are smaller

than the standard deviation of $\langle \epsilon_0(\boldsymbol{x}, t) \rangle_{\mathrm{VP},R}$.

It can be already concluded from Figs. 2, 7, A1 and A5 that $\epsilon_G$ is the most promising candidate to capture the true trend. However, to fully answer the above questions, we need to conduct more in depth analysis. The upper plot in Fig. 8 shows the re-scaled and coarse-grained dissipation field $\langle \epsilon_0(\boldsymbol{x}, t) \rangle_{\mathrm{VP},R}$ for a sliding window of size $R \approx 5500\eta_K$ and a turbulence intensity $I = 10\%$ obtained from track of one virtual probe for case DNS 2.0 ("probe 0"). Consistent with results shown

earlier, $\langle \epsilon_G \rangle_R$ follows $\langle \epsilon_0(\boldsymbol{x}, t) \rangle_{\mathrm{VP},R}$ best in comparison with $\langle \epsilon_{I2} \rangle_R$ and $\langle \epsilon_L \rangle_R$. Both $\langle \epsilon_{I2} \rangle_R$ and $\langle \epsilon_L \rangle_R$ are associated with substantial scatter, although $\langle \epsilon_{I2} \rangle_R$ has smaller deviations from the ground-truth overall. Other probe tracks sample different portions of the flow which is why a quantitative conclusion is not possible from one single probe. A more comprehensive evaluation of which method is able to capture the true trend is conducted below.

The lower plot in Fig. 8 shows $\langle \epsilon_{I2} \rangle_R$ together with the random error of $\epsilon_{I2}$ as defined by Eq. (25). Despite the strong

scatter, the ground-truth reference is nearly always within the errorbar of $\epsilon_{I2}$ with some exceptions, e.g. $r/\eta_K < 5000$ or $r/\eta_K \approx 44000$. It can also be seen that $\langle \epsilon_{I2} \rangle_R$ is, if at all, only weakly correlated with the ground-truth reference $\langle \epsilon_0(\boldsymbol{x}, t) \rangle_{\mathrm{VP},R}$ for a window size of $R/\eta_K \approx 5500$. This shows that it is extremely difficult, if at all possible, to track the true trend with low-resolution time records, which prevents the use of the gradient method.





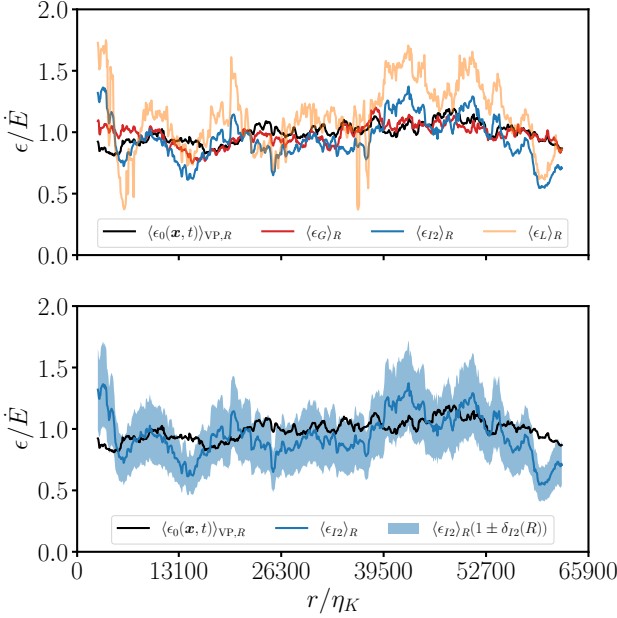

**Figure 8.** (Upper plot) Proof of concept for estimating the coarse-grained energy dissipation rate $\langle \epsilon_0(\boldsymbol{x},t) \rangle_{\mathrm{VP},R}$ re-scaled by the energy injection rate $\dot{E}$ via the one-dimensional surrogates $\langle \epsilon_G \rangle_R$, $\langle \epsilon_{I2} \rangle_R$, and $\langle \epsilon_L \rangle_R$ for $\mathrm{R}_\lambda = 142$, $R/\eta_K \approx 5500$, $\theta = 0°$ and a turbulence intensity $I = 10\%$ (DNS 2.0). All estimates are re-scaled by the energy injection rate $\dot{E}$, too. We narrowed the fit-range to $20\eta_K \leq r \leq 200\eta_K$ ensuring optimal fit results. (Lower plot) Comparison between $\langle \epsilon_{I2} \rangle_R / \dot{E}$ with estimated random error according to Eq. (25) for the averaging window $R$ and $\langle \epsilon_0(\boldsymbol{x},t) \rangle_{\mathrm{VP},R}$.

To assess this correlation more quantitatively, we evaluate Pearson's correlation coefficient between the ground-truth ref-
erence $\langle \epsilon_0(\boldsymbol{x},t) \rangle_{\mathrm{VP},R}$ and $\epsilon_G$, $\epsilon_{I2}$ as well as $\epsilon_L$, respectively, as a function of the re-scaled averaging window size $R/\eta_K$ for
all virtual probes of case DNS 2.0. As an example, Pearsons correlation coefficient between $\epsilon_0(\boldsymbol{x},t) \rangle_R$ and $\epsilon_{I2}$ is 0.33 in
Fig. 8 (upper plot). Figure 9A shows the ensemble averages of Pearson's correlation coefficient together with the standard
error (shaded area). While $\langle \epsilon_G \rangle_R$ has a pronounced correlation with the ground-truth reference $\langle \epsilon_0(\boldsymbol{x},t) \rangle_{\mathrm{VP},R}$, both $\langle \epsilon_{I2} \rangle_R$ and
$\langle \epsilon_L \rangle_R$ are only very weakly correlated with $\langle \epsilon_G \rangle_R$.

The effect of $\mathrm{R}_\lambda$ on Pearson's Correlation coefficient is shown in Fig. 9B also for the VDTT experiments at various $\mathrm{R}_\lambda$.
Here, we compare $\epsilon_{I2}$ and $\epsilon_L$ to $\epsilon_G$ in the absence of ground-truth. To ensure a negligible systematic error, we chose a fixed
averaging window of $R = 30L_{11}$ for each $\mathrm{R}_\lambda$. Figure 9B shows that the correlation for $\epsilon_{I2}$ is always higher than that of $\epsilon_L$
except for very low $\mathrm{R}_\lambda$. There is a non-monotonic behavior in the correlation coefficients in Fig. 9B that seems to be related
to the skewness values shown in Fig. A6. Nonetheless, there is a clear increase in correlation coefficients with $\mathrm{R}_\lambda$. Firstly, the
random error of $\delta_{I2}(R)$ ranges from 20% to 40% at $R = 30L_{11}$. Secondly, the kurtosis of the instantaneous energy dissipation
field scales with $\mathrm{R}_\lambda^{3/2}$ (Kolmogorov, 1962; Pope, 2000) which is why the variability in the instantaneous energy dissipation
field increases with $\mathrm{R}_\lambda$. Hence, at small $\mathrm{R}_\lambda^{3/2}$ and $R = 30L_{11}$, $\langle \epsilon_{I2} \rangle_{30L_{11}}$ scatters only randomly around the global mean energy
dissipation rate (with a 3% standard deviation of $\langle \epsilon_G \rangle_{30L_{11}}$), which is why the correlation coefficient is low. In contrast, at large





$R_\lambda$ and $R = 30L_{11}$, the locally averaged mean energy dissipation rate $\langle \epsilon_G \rangle_{30L_{11}}$ fluctuates stronger ($\approx 30\%$ standard deviation

of $\langle \epsilon_G \rangle_{30L_{11}}$) where $\delta_{I2}(R)$ is already comparable.

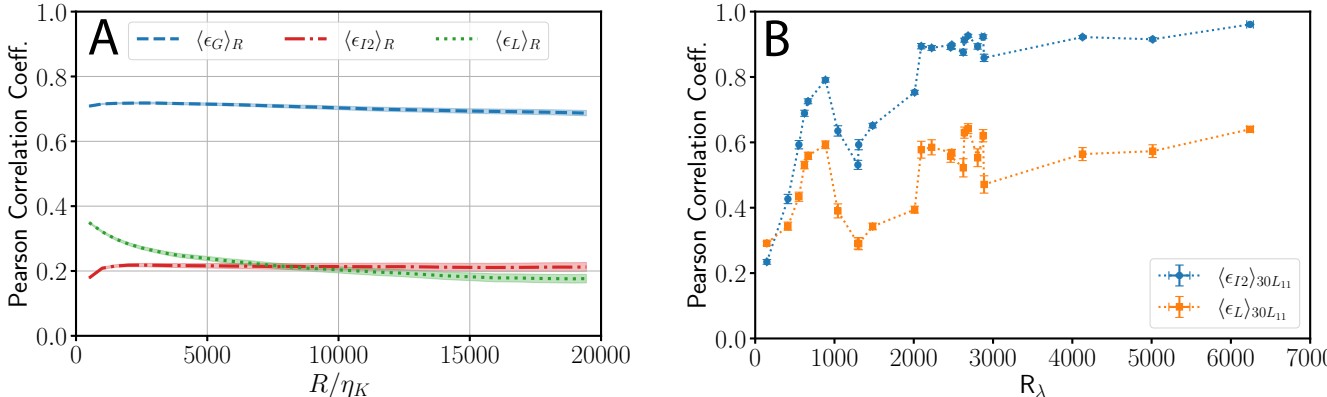

**Figure 9.** A: Dependence of the Pearson correlation coefficient between $\langle \epsilon_i \rangle_R$ and $\langle \epsilon_0(\boldsymbol{x},t) \rangle_{\mathrm{VP},R}$ as a function of the re-scaled averaging window $R/\eta_K$ where $i \in \{G, I2, L\}$. Time records of the longitudinal velocity by all virtual probes and $\langle \epsilon_0(\boldsymbol{x},t) \rangle_{\mathrm{VP},R}$ are taken from DNS 2.0 with $R_\lambda = 142$, turbulence intensity $I = 10\%$ and perfect alignment ($\theta = 0°$). The shaded region is given by the standard error. B: Dependence of the Pearson correlation coefficient between $\langle \epsilon_{I2,L} \rangle_R$ and $\langle \epsilon_G \rangle_R$ as a function $R_\lambda$ for a fixed re-scaled averaging window $R = 30L_{11}$. The error bars of the ensemble-averaged coefficients are given by the standard error.

## 4 Summary

We have presented an extensive review on the analysis procedure for estimating the energy dissipation rate from single-point one-dimensional velocity time records along with an overview of the advantages and disadvantages (see Table 1). The main methods considered in this study are the gradient method $\epsilon_G$, the 2nd-order SF (inertial range) method $\epsilon_{I2}$, the spectral method

$\epsilon_S$, and the scaling argument $\epsilon_L$. We have provided a systematic assessment of the accuracy of inferring the energy dissipation rate from such one-dimensional velocity time series as a function of turbulence intensity, probe orientation with respect to the longitudinal direction, and the effect of a finite averaging window size. We used DNS data with Reynolds numbers in the range $74 \leq R_\lambda \leq 321$ as well as experimental data from high-resolution one-dimensional wind tunnel measurements with Reynolds numbers in the range $147 \leq R_\lambda \leq 5864$ to evaluate the performance of different methods against robust benchmark values.

The results presented in this study help to assess the accuracy of the energy dissipation rate estimates as a function of several parameters, such as finite turbulence intensity, misalignment between sensor and longitudinal flow direction, and finite size of the averaging window. The main results are:

- Each method could reproduce the ground-truth reference $\langle \epsilon(\boldsymbol{x},t) \rangle$ to within less than 10 % for well converged statistics and at low turbulence intensity. The most accurate method is the gradient method ($\epsilon_G$) and the least accurate method is

the one based on the 4/5 law ($\epsilon_{I3}$) (see Figure 2). The "reference-compared" systematic error tend to be overestimated



due to the global choice of the fit-range, e.g. lower systematic errors for $\epsilon_{I2}$ can be obtained by choosing a fit-range for each DNS dataset that is in a range where the scaling of the structure function is closest to the expected scaling.

– In the case of finite turbulence intensities, $\epsilon_G$, $\epsilon_S$ and $\epsilon_{I2}$ systematically overestimate the ground-truth energy dissipation rate. The gradient method ($\epsilon_G$) is most affected by a finite turbulence intensity $I$ whereas $\epsilon_{I2}$ is the least affected (see Figs. 3 and 4B). The overestimation can be captured by a random advection model (Fig. 4). Regarding the small-scale estimate $\epsilon_G$, the error formula provided by Lumley (1965) ($\beta_G \propto 5I^2$) captures the effect of random advection.

– Considering the probe orientation, the gradient method ($\epsilon_G$) is most affected by misalignment between the probe orientation and the longitudinal flow direction whereas $\epsilon_{I2}$ is the least affected (Figure 5) (compare Eqs. (20), (18), and (19)).

– We provide scaling arguments $\delta_i(R)$ to estimate the required averaging window size optimized for a desired random error threshold for $\epsilon_G$, Eq. (30), $\epsilon_{I2}$, Eq. (29), and $\epsilon_L$, Eq. (29). With this, we can estimate a coarse-grained energy dissipation rate to within a predicted uncertainty as shown in Fig. 8. Systematic errors $\beta_i$ are smaller than random errors $\delta_i(R)$ for $R > 2L_{11}$.

– The random error of the gradient method $\delta_G(R)$ converges at least 4-5 times faster than $\epsilon_{I2}$ (compare Eqs. (29) and (30)).

– Only $\epsilon_G$ reliably estimates the transient energy dissipation rate $\langle \epsilon \rangle_R$ although it is most vulnerable to experimental imperfections/limitations.

*Code availability.* The code can be shared by the corresponding author upon request.

*Data availability.* The data can be shared by the corresponding author upon request.

*Author contributions.* TB, MW and EB provided the velocity data. MS, EB, MW, and GB conceptualized the study. MS developed, validated and ran the analysis code. The theoretical modeling was performed by MS, TB, and MW. MS, TB, MW, and GB analyzed and interpreted the data. MS, TB, MW, and GB wrote the initial draft. MS, TB, EB, MW, and GB proofread and edited the final manuscript.

*Competing interests.* The authors declare that they have no conflict of interest.

*Acknowledgements.* This work was supported by the Fraunhofer – Max-Planck Cooperation Program through the TWISTER project. MS was financially supported by the Konrad-Adenauer-Stiftung. TB was financially supported by a fellowship of the IMPRS for Physics of Biological and Complex Systems. We thank Christian Küchler and Gregory Bewley for providing us with VDTT data, and we thank David





Kleinhans, Christian Küchler, Freja Nordsiek, Naseem Ali and Holger Nobach for helpful discussions. Furthermore, we thank Cristian C. Lalescu and Bérenger Bramas for their support and development of the TurTLE code used in this study. We are grateful for the support by the MPIDS High Performance Computation (HPC) team for providing and maintaining computational resources.

## Appendix A: Preliminaries on Second-Order Statistics

As discussed in detail below, the mean energy dissipation rate can be related to second-order statistics of the velocity field,
either in terms of velocity gradients or in terms of velocity increments. In any case, the two-point velocity covariance tensor turns out to be the central quantity of interest, from which the second-order structure function tensor, the spectral energy tensor and the velocity gradient covariance tensor can be obtained.

In the following, we assume zero-mean SHI turbulence so that two-point quantities depend only on the separation vector $\boldsymbol{r}$, all averages are invariant under rotations of the coordinate system, and the mean squared velocity fluctuation is identical for all
velocity components, i.e. $\langle u'^2 \rangle = \langle u_1'^2 \rangle = \langle u_2'^2 \rangle = \langle u_3'^2 \rangle$. We provide an overview of the most relevant definitions, their notation and conventions. This section does not explicitly discuss the effect of the averaging window, but the definitions presented can be applied to windowed inputs with no or straightforward modifications.

Under the given assumptions, the two-point velocity covariance tensor takes the form (e.g. Pope, 2000; Robertson, 1940; Batchelor, 1953)

$$R_{ij}(\boldsymbol{r}) = \langle u_i'(\boldsymbol{x}+\boldsymbol{r},t)u_j'(\boldsymbol{x},t)\rangle = \langle u'^2 \rangle \left( g(r)\delta_{ij} + [f(r)-g(r)]\frac{r_i r_j}{r^2} \right), \tag{A1}$$

where $f(r) = R_{11}(r)/R_{11}(0)$ and $g(r) = f(r) + r\partial_r f(r)/2$ are the longitudinal and transverse autocorrelation functions, respectively, with $f(0) = g(0) = 1$. Notably, if one chooses $\boldsymbol{r} = r\boldsymbol{e}_1$, $R_{11}(r) = \langle u'^2 \rangle f(r)$, $R_{22}(r) = R_{33}(r) = \langle u'^2 \rangle g(r)$, and all other components vanish (e.g. Pope, 2000). As a remarkable consequence, $R_{ij}(\boldsymbol{r})$ is uniquely defined by $f(r)$ in isotropic turbulence. As mentioned below, the integral length scale as well as the Taylor microscale are determined by $f(r)$ (Pope, 2000).

Analogously, a covariance tensor can be defined for velocity increments, i.e. the second-order velocity structure function tensor (Pope, 2000; Davidson, 2015)

$$D_{ij}(\boldsymbol{r}) = \langle [u_i'(\boldsymbol{x}+\boldsymbol{r},t) - u_i'(\boldsymbol{x},t)] [u_j'(\boldsymbol{x}+\boldsymbol{r},t) - u_j'(\boldsymbol{x},t)]\rangle = D_{NN}(r)\delta_{ij} + [D_{LL}(r) - D_{NN}(r)]\frac{r_i r_j}{r^2}. \tag{A2}$$

The longitudinal second-order structure function $D_{11}(r)$ is related to $f(r)$ by (e.g. Pope, 2000; Davidson, 2015)

$$D_{11}(\boldsymbol{r} = r\boldsymbol{e}_1) = D_{LL}(r) = \langle (u_1'(\boldsymbol{x}+r\boldsymbol{e}_1,t) - u_1'(\boldsymbol{x},t))^2 \rangle = 2\langle u'^2 \rangle (1 - f(r)). \tag{A3}$$

As explained below, measuring the longitudinal second-order structure function $D_{LL}(r)$, the mean energy dissipation rate can be inferred from the inertial-range scaling of the longitudinal structure function (cf. Eq. (7)).





| Symbol | Definition | Equation | Dimensions |
|---|---|---|---|
| $A$ | large-scale anisotropy parameter | $3\langle u_1'^2\rangle/(2k)$ | |
| $C_K$ | Kolmogorov constant related to $E(\kappa)$ | 1.5 | |
| $C_\epsilon$ | dissipation constant | 0.5 | |
| $D_{ij}(\boldsymbol{r})$ | second-order velocity structure function tensor | (A2) | $L^2T^{-2}$ |
| $E(\kappa)$ | energy spectrum function | $\iiint_{-\infty}^{\infty} \frac{1}{2}\Phi_{ii}(\boldsymbol{\kappa})\delta(|\boldsymbol{\kappa}|-\kappa)\mathrm{d}\boldsymbol{\kappa}$ | $L^3T^{-2}$ |
| $E_{11}(\kappa_1)$ | one-dimensional energy spectrum | (A7) | $L^3T^{-2}$ |
| $F_{11}(f)$ | power spectral density of longitudinal velocity | $\frac{\Delta t}{N}\mathcal{F}(u_1(t))\mathcal{F}^*(u_1(t))$ | $L^2T^{-1}$ |
| $\mathcal{F}(x)$ | (discrete) Fourier transform | $\sum_{j=0}^{N-1} x(t_j)\exp(-2\pi i t_j/\Delta t)$ | |
| $I$ | turbulence intensity | $\sigma_{u_1'}/U$ | |
| $L$ | length scale characteristic of large eddies, e.g. energy injection scale | $k^{3/2}/\epsilon$ | $L$ |
| $L_{11}$ | longitudinal integral length scale of the turbulent flow | (11) | $L$ |
| $R_{ij}(\boldsymbol{r})$ | velocity (two-point, one-time velocity auto-) covariance tensor | (A1) | $L^2T^{-2}$ |
| $R_{ijkl}(\boldsymbol{r})$ | velocity gradient covariance tensor | (A4) | $T^{-2}$ |
| Re | Reynolds number | $\frac{UL}{\nu}$ | |
| $\mathrm{R}_\lambda$ | Taylor-scale Reynolds number | $\sqrt{\frac{15\sigma_u^4}{\nu\langle\epsilon\rangle}}$ | |
| $\mathrm{Re}_L$ | turbulence Reynolds number | $\frac{k^{1/2}L}{\nu}$ | |
| $S_{ij}$ | strain rate tensor | $(\frac{\partial u_i(\boldsymbol{x},t)}{\partial x_j}+\frac{\partial u_j(\boldsymbol{x},t)}{\partial x_i})/2$ | $T^{-1}$ |
| $T$ | longitudinal integral time scale of the turbulent flow | $\int_0^\infty f(\tau)\mathrm{d}\tau$ | $T$ |
| $\boldsymbol{U}, U_\tau$ | global-mean velocity vector of the flow and the local-mean of the longitudinal velocity component for averaging window of duration $\tau$ relative to the virtual probe | (1) | $LT^{-1}$ |
| $a_{\mathrm{fit}}$ | fit parameter related to Eq. (29) | | |
| $\dot{E}$ | energy injection rate of the DNS | | $L^2/T^{-3}$ |
| $f$ | frequency | | $T^{-1}$ |
| $f(r)$ | longitudinal velocity auto-correlation [coefficient] function | $R_{11}(r)/R_{11}(0)$ | |
| $g(r)$ | transverse velocity auto-correlation [coefficient] function | $f(r)+r\partial_r f(r)/2$ | |
| $k$ | turbulent kinetic energy | $(u_1'^2+u_2'^2+u_3'^2)/2$ | $L^2T^{-2}$ |
| $\boldsymbol{r}, r$ | distance vector (or rather radial coordinate) and its absolute value | | $L$ |
| $s_{ij}$ | [velocity] fluctuation strain rate tensor | $(\frac{\partial u_i'(\boldsymbol{x},t)}{\partial x_j}+\frac{\partial u_j'(\boldsymbol{x},t)}{\partial x_i})/2$ | $T^{-1}$ |
| $t$ | time | | $T$ |
| $\boldsymbol{u}$ | (Eulerian) velocity vector of the flow | $u_1\boldsymbol{e}_1+u_2\boldsymbol{e}_2+u_3\boldsymbol{e}_3$ | $LT^{-1}$ |
| $\boldsymbol{u}'$ | velocity fluctuation vector of the flow | $\boldsymbol{u}-\boldsymbol{U}$ | $LT^{-1}$ |
| $\langle u_1'^2\rangle$ | variance of longitudinal velocity fluctuations | $\int_0^\infty E_{11}(\kappa_1)\mathrm{d}\kappa_1$ | $L^2T^{-2}$ |
| $\boldsymbol{x}$ | position vector | $x_1\boldsymbol{e}_1+x_2\boldsymbol{e}_2+x_3\boldsymbol{e}_3$ | $L$ |
| $\Phi_{ij}(\boldsymbol{\kappa})$ | energy tensor (velocity spectrum tensor) | (A5) | $L^5T^{-2}$ |
| $\alpha$ | fit parameter related to Eq. (29) | | |
| $\delta_{ij}$ | Kronecker delta | | |
| $\Delta t$ | time increment | $\min\{t_{j+1}-tj\}$ | $T$ |
| $\epsilon$ | energy dissipation rate | | $L^2T^{-3}$ |
| $\epsilon_0(\boldsymbol{x},t)$ | instantaneous energy dissipation rate | (3) | $L^2T^{-3}$ |
| $\epsilon_R$ | locally volume averaged energy dissipation rate | (4) | $L^2T^{-3}$ |
| $\langle\epsilon\rangle$ | global-mean energy dissipation rate (rate of dissipation of turbulent kinetic energy) | (5) | $L^2T^{-3}$ |
| $\epsilon_{ijk}$ | Levi-Cevita tensor | | |
| $\zeta_n$ | $n$th-order structure function exponent | $\frac{\mathrm{d}\log D_{L\ldots}(r)}{\mathrm{d}\log r}$ | |
| $\eta_K$ | Kolmogorov length scale | $(\nu^3/\langle\epsilon\rangle)^{1/4}$ | $L$ |
| $\theta$ | angle of incidence between probe orientation and longitudinal flow direction | | $°$ |
| $\boldsymbol{\kappa}$ | wave vector | | $L$ |
| $\lambda$ | longitudinal Taylor (micro-)scale | $\sqrt{\frac{30\nu u_1'^2}{\langle\epsilon\rangle}}$ | $L$ |
| $\nu$ | kinematic viscosity | | $L^2T^{-1}$ |
| $\sigma_x$ | standard deviation of quantity $x$ | | |
| $\sigma_{u_1'}$ | root mean square of longitudinal velocity fluctuations | | $LT^{-1}$ |
| $\omega$ | angular frequency | $2\pi f$ | $T$ |
| $\langle\ldots\rangle_N$ | ensemble average | | |
| $\langle\ldots\rangle_R$ | volume average [line average for 1D signal] | | |

**Table A1.** Nomenclature for the turbulent flow. If our naming convention differs from the terminology in (Pope, 2000), we add the convention of Pope in parentheses. Equations are either directly given or referenced from definitions below.





| Symbol | Definition |
|---|---|
| 1, 2, 3 | indices of vectors and tensors |
| $C$ | cutoff |
| $D$ | dissipation range |
| $G$ | gradient |
| $I2$ | inertial range of second-order structure function |
| $I3$ | inertial range of third-order structure function |
| $L$ | longitudinal |
| $N$ | ensemble (e.g. $\langle \cdot \rangle_N$ for ensemble average) |
| $N\ldots$ | transverse (e.g. $NN$ for transverse second-order structure function) |
| $R$ | averaging window size in space |
| $S$ | inertial range of the power spectral density |
| $p$ | probe |
| ref | (ground-truth) reference |
| VP | virtual probe |
| $\tau$ | averaging window size in time |

**Table A2.** Nomenclature for the subscripts.

Furthermore, the velocity gradient covariance tensor can also be defined in terms of the velocity covariance tensor

$$R_{ijkl}(\boldsymbol{r}) = \left\langle \frac{\partial u'_i(\boldsymbol{x},t)}{\partial x_k} \frac{\partial u'_j(\boldsymbol{x},t)}{\partial x_l} \right\rangle = -\lim_{r \to 0} \partial_{r_k} \partial_{r_l} R_{ij}(\boldsymbol{r}) \,. \tag{A4}$$

Since the local and instantaneous energy dissipation rate (cf. Eq. (3)) is defined in terms of the strain rate tensor $S_{ik} =$
$(\partial u'_i(\boldsymbol{x},t)/\partial x_k + \partial u'_k(\boldsymbol{x},t)/\partial x_i)/2$, the mean energy dissipation rate can be directly related to contractions of the velocity gradient covariance tensor. Note that in a turbulent flow with zero-mean velocity, the strain rate tensor $S_{ik}$ equals the fluctuation strain rate tensor $s_{ik}$.

The two-point velocity covariance tensor can be expressed in Fourier space through the spectral energy tensor (Pope, 2000)

$$\Phi_{ij}(\boldsymbol{\kappa}) = \frac{1}{(2\pi)^3} \iiint\limits_{-\infty}^{+\infty} R_{ij}(\boldsymbol{r}) e^{-i\boldsymbol{\kappa}\cdot\boldsymbol{r}} \mathrm{d}\boldsymbol{r} \,, \tag{A5}$$

where $\boldsymbol{\kappa}$ is the wave vector. For SHI turbulence, $\Phi_{ij}(\boldsymbol{\kappa})$ takes the form

$$\Phi_{ij}(\boldsymbol{\kappa}) = \frac{E(\kappa)}{4\pi\kappa^2} \left( \delta_{ij} - \frac{\kappa_i \kappa_j}{\kappa^2} \right) \tag{A6}$$

where $E(\kappa)$ is the energy spectrum function.

Since access to the full energy spectrum function is not always available, one-dimensional spectra are of interest, too. The mean energy dissipation rate can be estimated from the inertial range scaling of the longitudinal one-dimensional spectrum (as
shown in Eq. (9)), which can be calculated by both the energy spectrum function and the velocity covariance tensor (Pope,





2000)

$$E_{11}(\kappa_1) = \int\limits_{\kappa_1}^{\infty} \frac{E(\kappa)}{\kappa} \left(1 - \frac{\kappa_1^2}{\kappa^2}\right) \mathrm{d}\kappa = \frac{1}{\pi} \int\limits_{-\infty}^{\infty} R_{11}(\boldsymbol{e}_1 r_1) e^{-i\kappa_1 r_1} \mathrm{d}r_1 \,, \tag{A7}$$

with the wavenumber $\kappa_1$ corresponding to the $\boldsymbol{e}_1$-direction and $R_{11}(0) = \langle u'^2 \rangle = \int_0^{\infty} E_{11}(\kappa_1)\mathrm{d}\kappa_1$.

This concludes the second-order statistics in terms of the velocity that we consider in the following to determine the mean
energy dissipation rate.

## Appendix B: Impact of random sweeping effects on gradient method

In the following, we illustrate how one obtains the expression in terms of the turbulence intensity for the impact of random
sweeping effects on the dissipation rate estimate using the gradient method $\epsilon_G = \langle \epsilon \rangle [1 + 5I^2]$ Lumley (1965); Wyngaard and
Clifford (1977). We consider a model wavenumber-frequency spectrum (Wilczek and Narita, 2012; Wilczek et al., 2014),
which is based on the same modeling assumptions used in Wyngaard and Clifford (1977). It enables us to conduct a systematic
assessment of the interplay between Taylor's hypothesis and the random sweeping effects. The model wavenumber-frequency
spectrum tensor $\Phi_{ij}(\boldsymbol{\kappa}, \omega)$ can be derived from an elementary linear random advection model (Kraichnan, 1964; Wilczek and
Narita, 2012; Wilczek et al., 2014), which in case of SHI turbulence can be expressed in terms of the energy spectrum tensor
$\Phi_{ij}(\boldsymbol{\kappa})$:

$$\Phi_{ij}(\boldsymbol{\kappa}, \omega) = \frac{\Phi_{ij}(\boldsymbol{\kappa})}{\sqrt{2\pi\kappa^2 I^2 U^2}} \exp\left(-\frac{(\omega/U - \kappa_1)^2}{2\kappa^2 I^2}\right). \tag{B1}$$

Within the model, the wavenumber-frequency spectrum $\Phi_{ij}(\boldsymbol{\kappa}, \omega)$ consists of the energy spectrum tensor in wavenumber space
$\Phi_{ij}(\boldsymbol{\kappa})$ multiplied by a Gaussian frequency distribution. $\Phi_{ij}(\boldsymbol{\kappa}, \omega)$ has a mean value determined by $\omega = \kappa_1 U$, i.e. Taylor's hy-
pothesis expressed in Fourier space, and a variance proportional to the turbulence intensity. When the turbulence intensity tends
to zero at fixed mean velocity, the frequency distribution tends to a delta function, re-establishing the one-to-one correspon-
dence between the frequency and the wavenumber in the direction of the mean flow. To establish the connection to the different
methods using longitudinal components and Taylor's hypothesis, we consider the $i = j = 1$ component of Eq. (B1). One ob-
tains the estimate for the longitudinal wavenumber spectrum based on Taylor's hypothesis, which includes the effect of random
sweeping, by first integrating over the wavevector space. This leads to the frequency spectrum $\tilde{F}_{11}(\omega)$, which corresponds to
the one obtained from temporal single-point measurements of the longitudinal velocity component. Then, one applies Taylor's
hypothesis, corresponding to the substitution $\omega = \kappa_1 U$ which leads to

$$E_{11}(\kappa_1)d\kappa_1 \stackrel{\omega=\kappa_1 U}{=} \frac{1}{U}\tilde{F}_{11}(\omega)d\omega = \frac{2}{U}\left[\int \Phi_{11}(\boldsymbol{\kappa}', \omega)d\boldsymbol{\kappa}'\right]d\omega\,. \tag{B2}$$





Finally, this enables us to evaluate the influence of random sweeping on the gradient method since it is closely related to the wavenumber spectrum. Expressed in wavenumber space, the relation, Eq. (6), takes the following form, where we insert Eqs. (B1)-(B2) and solve the corresponding Gaussian integral over $\omega$ in the second step:

$$\epsilon_G = 15\nu \int \kappa_1^2 E_{11}(\kappa_1)d\kappa_1 = \langle\epsilon\rangle[1 + 5I^2]. \tag{B3}$$

Hence, we recover, as expected, the result by Lumley (1965) and Wyngaard and Clifford (1977).

To numerically assess how random sweeping smears out the spectrum at finite turbulence intensities (see Fig. 4), we assumed a model wavenumber spectrum, (Pope, 2000, Eq. 6.246 ff.):

$$E(\kappa) = C_K \langle\epsilon\rangle^{2/3} \kappa^{-5/3} \left(\frac{\kappa L}{[(\kappa L)^2 + c_L]^{1/2}}\right)^{5/3+p_0} \exp(-\beta\kappa\eta_K), \tag{B4}$$

where $L$ is the energy injection scale and $c_L = 6.78$, $p_0 = 2$ and $\beta = 2.094$ are positive constants. Based on this model wavenumber spectrum we first obtain $E_{11}(\kappa)$ through Eq. (A7). Where one can then evaluate Eq. (B2) resulting in the spectrum smeared out by random sweeping.

## Appendix C: Effect of Probe Misalignment

Here, we derive estimates for the systematic error due to misalignment between the (virtual) anemometer and the global mean wind direction $\frac{U}{|U|}$. This misalignment is captured by an angle $\theta$, which is assumed to be constant in time and space. In general, the rotation matrix around an arbitrary rotation axis $\hat{n}$ with $n_i n_i = 1$ is given by $\mathcal{R}_{ij}^{\hat{n}}(\theta) = (1 - \cos\theta)n_i n_j + \cos\theta\delta_{ij} + \sin\theta\epsilon_{ijk}n_k$ (e.g. Cole, 2015; Hanson, 2011), where $\epsilon_{ijk}$ is the Levi-Cevita tensor and $\delta_{ij}$ the Kronecker delta. At first, we consider the covariance tensor $R_{ij}(\boldsymbol{r}')$ as the integral length scale, the second-order structure function tensor as well as the velocity gradient covariance tensor depend on $R_{ij}(\boldsymbol{r}')$ (Eqs. (A1), (A2) and (A4), respectively). In the sensor frame of reference, the covariance tensor is given by

$$R_{ij}(\boldsymbol{r}') = \langle u'^2\rangle \left(g(r')\delta_{ij} + [f(r') - g(r')]\frac{r_i' r_j'}{r'^2}\right), \tag{C1}$$

where $r_i' = \mathcal{R}_{ij}^{\hat{n}}(\theta)r_j$ and $r' = r$. As only the longitudinal component in the sensor frame of reference is measured, Eq. (C1) reads for $i = j = 1$ and $\boldsymbol{r} = r\boldsymbol{e}_1$

$$R_{11}(\boldsymbol{r}') = \langle u'^2\rangle \left(g(r) + [f(r) - g(r)]\frac{\mathcal{R}_{1l}^{\hat{n}}(\theta)r_l \mathcal{R}_{1k}^{\hat{n}}(\theta)r_k}{r^2}\right) \tag{C2}$$

$$= \langle u'^2\rangle \left(g(r) + [f(r) - g(r)]\left(\cos^2\theta + n_1^4(1 - \cos\theta)^2 + 2n_1^2(1 - \cos\theta)\cos\theta\right)\right). \tag{C3}$$





For further simplification, we assume without loss of generality that the mean wind changes direction only in the horizontal plane. With this we can set $\hat{n} = 1e_3$, which yields for $r' = r'e_1'$

$$R_{11}(r'e_1')/\langle u'^2 \rangle = f(r') = \langle u'^2 \rangle \left( g(r) + [f(r) - g(r)] \cos^2 \theta \right) , \tag{C4}$$

which we interpret as the measured autocorrelation function. Then, the measured longitudinal integral length scale, Eq. (11), amounts to

$$L_{11}'(\theta) = \int_0^\infty dr' \, f(r') = \int_0^\infty dr \cos\theta(\cos^2\theta f(r) + (1 - \cos^2\theta)g(r)) = \frac{1}{2} L_{11} \cos\theta \left(1 + \cos^2\theta\right) , \tag{C5}$$

where the integration of $f(r)$ and $g(r)$ is carried out in the last step, see Eq. (11), while considering the fact that $L_{22} = L_{11}/2$ for isotropic turbulence (Pope, 2000).

An analogous argument also holds for the second-order structure function tensor, Eq. (A2):

$$D_{11}(r') = D_{NN}(r) + [D_{LL}(r) - D_{NN}(r)] \cos^2\theta = D_{LL}(r) \left( \frac{4 - \cos^2\theta}{3} \right) , \tag{C6}$$

where the transverse second-order structure function $D_{NN}(r) = D_{22}(r) = D_{33}(r)$ is expressed as $D_{NN}(r) = 4D_{LL}(r)/3 = 4C_2(r\epsilon)^{2/3}/3$ in SHI turbulence (Pope, 2000).

The misalignment error for the gradient method can be estimated analytically starting from the longitudinal component of the velocity gradient covariance tensor $R_{1111}$ — it can be also expressed in terms of the velocity covariance tensor Eq. (A4). Following similar arguments as above and starting from Eq. (A4), assuming $r = re_1$ and applying the rotation about an axis $\hat{n}$ with $n_i n_i = 1$, we obtain

$$R_{1111}(0) = -\lim_{r' \to 0} \partial_{r'} \partial_{r'} R_{11}(r'e_1') = -\frac{u'^2}{\cos^2\theta} \lim_{r \to 0} \partial_r^2 \left[ g(r) + [f(r) - g(r)] \frac{r^2 \cos^2\theta}{r^2} \right] , \tag{C7}$$

where $\partial_{r'} = \partial_r / \cos\theta$ due to the rotation. Using $\partial_r^2 g(r) = 2\partial_r^2 f(r) + \frac{r}{2}\partial_r^3 f(r)$ (Pope, 2000), the velocity gradient covariance tensor reduces to

$$R_{1111}(0) = -\frac{u'^2}{\cos^2\theta} \lim_{r \to 0} \left[ (2 - \cos^2\theta)\partial_r^2 f(r) + (1 - \cos^2\theta)\frac{r}{2}\partial_r^3 f(r) \right] \tag{C8}$$

$$= \left\langle \left( \frac{\partial u}{\partial x_1} \right)^2 \right\rangle \frac{2 - \cos^2\theta}{\cos^2\theta} , \tag{C9}$$

where $-u'^2 \lim_{r \to 0} \partial_r^2 f(r) = \langle (\partial u/\partial x_1)^2 \rangle$ (Pope, 2000) is used for the last step.



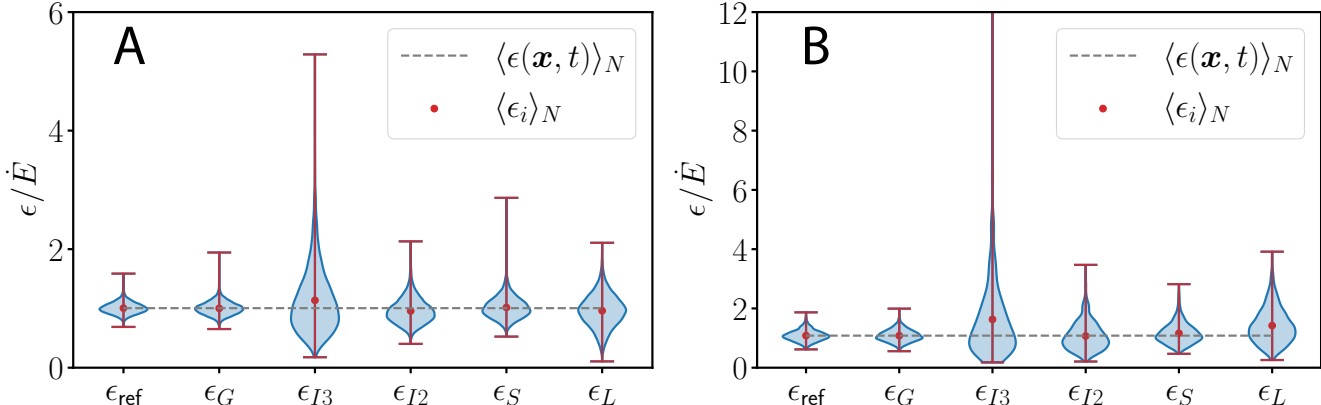

**Figure A1.** Validation of estimating the energy dissipation rate from $\epsilon_G, \epsilon_{I2}, \epsilon_{I3}, \epsilon_S$, and $\epsilon_L$. All estimates are re-scaled by the energy injection rate $\dot{E}$. The data is taken from DNS 1.1 (A) and 2.1 (B), turbulence intensity $I = 1\%$. The ensemble mean of each method $\langle \epsilon_i \rangle_N$ is denoted by red dots where the whiskers extend from the minimal to maximal estimate of $\epsilon_i$ where $i \in \{G, I3, I2, S, L\}$. As the inertial range of DNS 1.1 ($I = 1\%$, $\theta = 0°$ and maximal available averaging window) is not well pronounced due to the low $R_\lambda \sim 74$, we used the maximum of Eq. (7) in order to retrieve $\epsilon_{I2,3}$. The dashed line represents the global mean energy dissipation rate of DNS 1.1 and 2.1 ($R_\lambda = 219$, $I = 1\%$, $\theta = 0°$ and maximal available averaging window), respectively, which is approximated by the ensemble average of the true mean energy dissipation rate along the trajectory of each virtual probe. $\epsilon_{\mathrm{ref}}$ is the reference distribution of ground-truth global mean energy dissipation field originating from the dissipation field along the trajectory of each virtual probe.

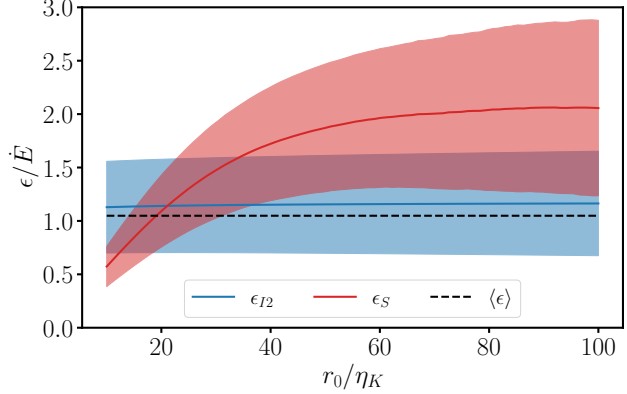

**Figure A2.** Estimates of the mean energy dissipation rate as a function of the fit-range for $\epsilon_{I2}$ and $\epsilon_S$ for DNS 3.1 (1000 probes, $R_\lambda = 302$, $I = 1\%$, $\theta = 0°$ and maximal available averaging window) re-scaled by the energy injection rate $\dot{E}$. The solid line represents the ensemble average whereas the shaded region is given by the standard deviation. $r_0 \in [10\eta_K, 100\eta_K]$ is the lower boundary of the fit-range for $\epsilon_{I2}$ where the upper boundary is fixed at $r_1 = 500\eta_K$. For $\epsilon_S$, the fit-range is given by $f \in [U/r_1, U/r_0]$. The dashed line denotes the global mean energy dissipation rate.



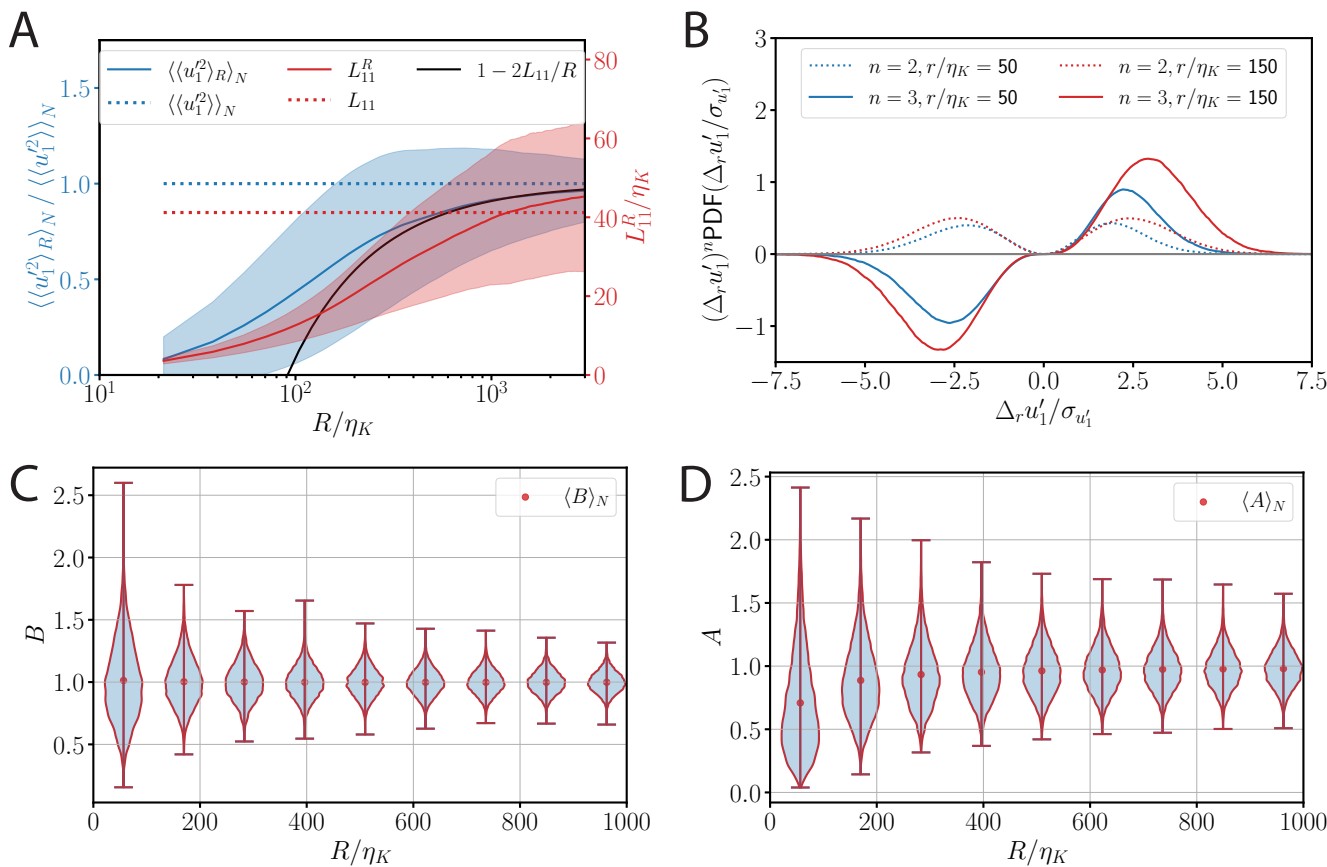

**Figure A3.** Convergence of higher-order statistical quantities and longitudinal integral length scale as well as small and large-scale anisotropy obtained from all virtual probes of DNS 1.1 ($R_\lambda = 74$, $I = 1\%$, $\theta = 0°$). (A) The variance of the longitudinal velocity fluctuations $\langle u_1'^2 \rangle$ and the longitudinal integral length scale $L_{11}^R$ as a function of averaging window size $R$ normalized by the Kolmogorov length scale $\eta_K$. $\langle u_1'^2 \rangle$ is defined by Eq. (21) and re-scaled by the ensemble-averaged variance of the longitudinal velocity fluctuations. For large $R$, $\langle\langle u_1'^2 \rangle_R \rangle_N$ converges to $\langle u_1'^2 \rangle \approx \langle\langle u_1'^2 \rangle\rangle_N$ (blue-dotted line) and the systematic error of the variance (solid-black line), Eq. (21), decays to 0. $L_{11}^R$ is the longitudinal integral length scale obtained from averaging windows of size $R$. For large $R$, $L_{11}^R$ should converge to $L_{11}$ (red-dotted line) which is not fully achieved in this range of $R$. (B) Premultiplied PDFs of second and third-order velocity increments over distances $r = 50\eta_K$ and $r = 150\eta_K$. The tails of the pre-multiplied PDFs have decayed to zero for large (and re-scaled) increments $\Delta_r u_1'/\sigma_{u_1'}$ so that they can globally considered to be converged. (C) Small-scale anisotropy based on the ratio of longitudinal gradients to the instantaneous energy dissipation $B = \epsilon_G/\langle\epsilon_0(\boldsymbol{x},t)\rangle$. In isotropic turbulence, $B = 1$ in average. (D) Large-scale anistropy parameter $A = 3\langle u_1'^2 \rangle/(2k)$ as a function of averaging window $R$ where $k$ is the turbulent kinetic energy and $\langle u_1'^2 \rangle$ the variance of the longitudinal velocity fluctuations. In isotropic turbulence, $A = 1$ in average.

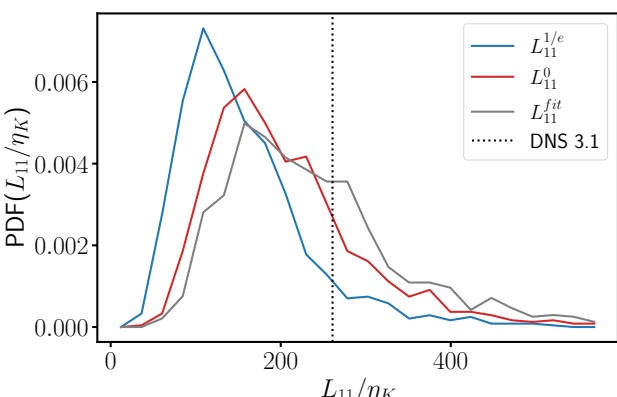

**Figure A4.** Different estimates of the integral length from DNS 3.1 with $R_\lambda = 302$, $I = 1\%$, $\theta = 0°$ and maximal available averaging window. $L_{11}^0$ is inferred from integrating $f(r)$ to its first zero whereas $L_{11}^{1/e}$ refers to the integration of $f(r) > 1/e$. $L_{11}^{fit}$ extends $f(r)$ with an exponential tail where the integration is performed up to infinity. The black dotted line is the reference from DNS 3.1 obtained by Eq. (11). All estimates are re-scaled by $\eta_K$.

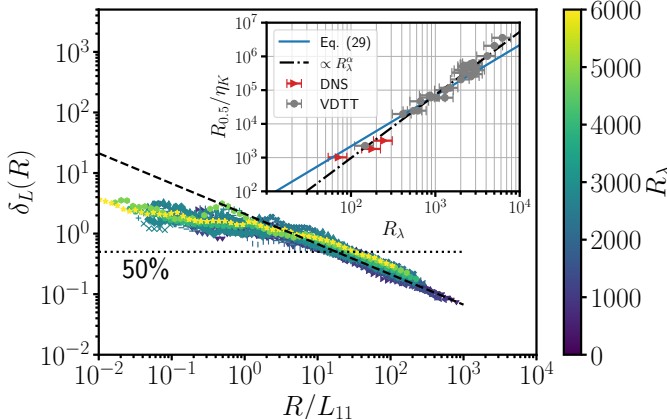

**Figure A5.** Convergence of energy dissipation rate estimates for $\epsilon_L$. The empirical random error $\delta_L(R)$ is plotted as a function of re-scaled averaging window size $R/L_{11}$ from VDTT experiments at various $R_\lambda$. The analytical result for the random error (Eq. (23)) is shown by the dashed black line. (Inset) The insets show the length of the averaging window in terms of $\eta_K$ where $\hat{\delta}_L^R$ is less than 50% as a function of Taylor microscale Reynolds number $R_\lambda$. The inset plot shows data from DNS 1.3, 2.3 and 3.3 (red triangles) and the VDTT (grey circle). The three red dots mark the experiments with the highest $R_\lambda$ where the isotropy of the grid forcing in the VDTT is not guaranteed anymore. The solid, blue line shows Eq. (29) resolved for $R_{0.5}/\eta_K$. The double logarithmic fit ($\log R/\eta_K = \frac{3}{4} \log \frac{9}{4} \frac{3}{20} - 2 \log a_{\text{fit}} + \alpha \log R_\lambda$) is performed for the scaling argument resulting in $\alpha = 1.87 \pm 0.06$ and $a_{\text{fit}} = 1.75 \pm 0.22$ (black dash-dotted line).





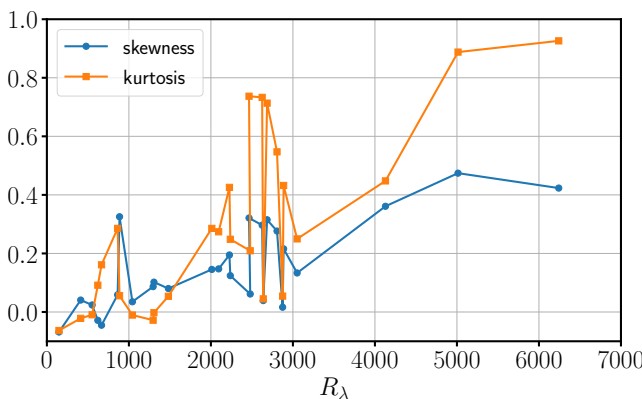

**Figure A6.** Skewness and kurtosis of all VDTT experiments as a function of $R_\lambda$. The skewness vanishes for normally distributed velocity time records. Similarly, the kurtosis equals 0 for normally distributed velocity time records, according to Fisher's convention.

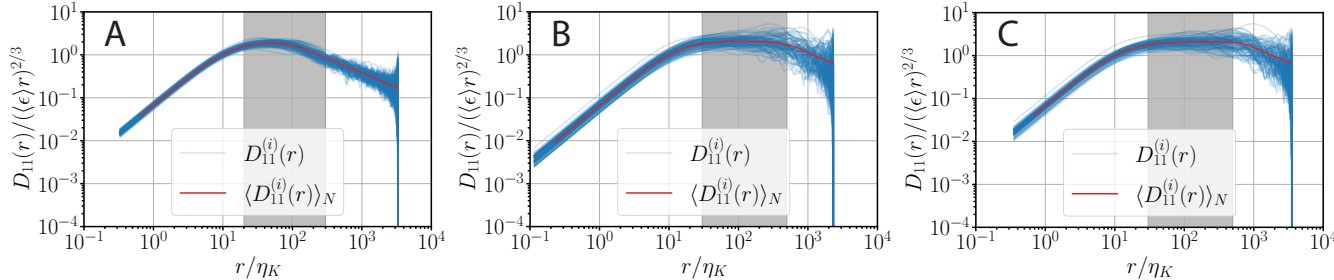

**Figure A7.** Compensated longitudinal second-order structure functions for DNS 1.1 (A), DNS 2.1 (B), and DNS 3.1 (C). The grey shaded region represents the fit-range (Eq. (7)) for each DNS. The individual longitudinal second-order structure functions are calculated from the velocity time records along the $e_1$-direction of each virtual probe (blue lines). The ensemble-averaged longitudinal second-order structure functions are shown in red.





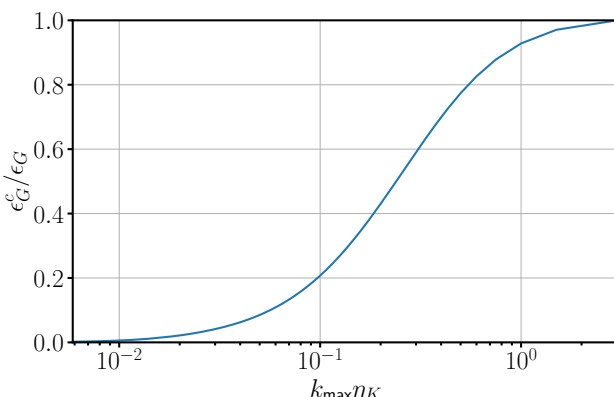

**Figure A8.** Resolution effect on $\epsilon_G$. $\epsilon_G^c$ refers to the coarse-grained velocity time record. Coarse-graining is realized by taking only every $n$th value of the fully resolved velocity time record where $n \in [1, 512]$, thereby controlling the resolution $k_{\max}\eta_K$. Velocity data are taken from DNS 2.0 ($R_\lambda = 142$, $R \approx 32000\eta_K$, $I = 10\%$, $\theta = 0°$).



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
