# Peer review of "Estimating the turbulent kinetic energy dissipation rate from one-dimensional velocity measurements in time"

_Atmospheric Measurement Techniques, 2023_

## Referee Comment (RC1)

**Estimating the turbulent kinetic energy dissipation rate from one-dimensional velocity measurements in time**

by: Marcel Schröder, Tobias Bätge, Eberhard Bodenschatz, Michael Wilczek, and Gholamhossein Bagheri

This paper addresses the fundamental problem of estimating "local" energy losses from one-dimensional velocity measurements as a function of time, with a focus on atmospheric conditions. Such observations can be made, for example, with single-component hot-wire anemometers and are critical to a better understanding of turbulent processes in the atmosphere. Since there are different methods and approaches to determine energy dissipation rates, there is also no doubt about the need to discuss in detail the methods and the sources of error associated with the measurement which has been done in great detail in this work.

Furthermore, there is also no doubt that an in-depth treatment of this topic is still missing in the literature and thus this work makes a very valuable contribution to atmospheric turbulence measurements and their accuracy. I can only congratulate the authors on this contribution and really have only one but important comment, which relates to its potential usefulness to a wider readership and should be easy to incorporate. I must also emphasize in this context that my background is rather experimental and I have not followed all derivations in detail. Therefore, as I said before, my main criticism also lies somewhat in the potential applicability of the material for experimenters. The discussions are for the most part very theoretical which is certainly absolutely okay, but from my perspective the applicability of the results suffers as a result. I think, especially in the summary, more clear recommendations for action could be formulated as to what is the most robust method for determining dissipation rates under given circumstances. My fear is that in future publications on experimental determinations of dissipation, this manuscript will be cited but rather generally and without direct application and that would be a pity because this manuscript definitely has the potential to serve as a reference for future measurements of dissipation.

After consideration of my comments, I unreservedly recommend the paper for publication. When I ask here for "major revisions", this does not refer to the scientific content but to the request to revise the paper for better applicability and partly also readability.

**Major comment (partly overlap with the introduction):**

The derivations and subsequent discussions of the various error contributions in the determination of the dissipation rate are described in great detail and quite formally - this is also absolutely fine and understandable, although the abundance of details reminds me partly more of a dissertation than a manuscript for AMT; however, you have chosen a meteorologically oriented, metrological journal with a corresponding audience. For this reason, I wonder how you can possibly get even more use for

atmospheric measurements from this really great material.  You already shifted a lot of details into the appendix which is highly appreciated but I feel that at several places there is still too much detail not necessary for estimating the uncertainty of dissipation. One prominent example in this context is the description of the active grid of your wind tunnel (in particular Eq. 12), this is very technical and does not contribute to the main topic at all - a reference would be more than sufficient. The same with some tables such as Tab 2 for example - do I really need all the details of the DNS?

I think too much detail can easily distract from the main topic and slow down the reading flow, and I suggest going through the manuscript carefully and - where possible - shortening it some more (although I know from own experience how hard and difficult shortening can be…).

A second way to make the material more valuable to a wider audience would be to include some sort of recommendation in the summary section. Such a brief discussion would be very helpful for application-oriented colleagues.

**Specific/minor comments:**

Abstract: I think it is not common to define abbreviations in the abstract if they are not used again in the abstract itself - I suggest avoiding it.

**Introduction:**

The introduction is general nicely and clearly written  with only a few places where I suggest some more details and information:

Line 29ff: Maybe a short explanation what exactly is meant by "instantaneous energy dissipation field" is helpful at this place; what are the "high spatial/temporal scales" mentioned in line 29? Please specify!

line 32: if the averaging is over the instantaneous energy dissipation rate, why is the index "0" missing here?

line 36ff: can you please provide an example why the locally averaged dissipation is of importance (although I completely agree with this statement)?

line 49ff: this is probably true, but at least airborne turbulence observations with a sufficient high true airspeed yielding a low turbulence intensity where applying Taylor's hypothesis should be fine - right? So your comment is more related to ground-based observations or tethered systems (balloons, kites) combined with high turbulence intensities – should be mentioned.

**Chapter 2**

I don't understand why the strain rate tensor in Eq 3 is in capitols but in tab 1 not – please specify.

Title of Sec 2.2.3: not sure about how the subtitle compares to the subtitle of subsection 2.2.2. The spectral method is also an "indirect method" right? So, I think the subtitles should be similar and only differ for "spectral" and "structure function"?

Eq 11: maybe I missed it, but the autocorrelation function $f(r)$ has not been defined/introduced yet – right?

Fig 1A and C: Symbols are partly poorly resolved and pixelated and therefore difficult to read.

line 261: please provide a reference for the given number range for $R_\lambda$ under atmospheric conditions

line 262: maybe for the broader audience one or two sentences should be included about the basic motivation using SF6 at this high pressure and why not working at atmospheric conditions. I think the most of the readers do not know about the advantage of SF6 and what type of gas it is.

I'm not sure Eq 12 is necessary to understand what the experiment is about; the equation formalizes the text unnecessarily and a description in the text is more helpful and sufficient

Line 270: Why using three setups with quite comparable Taylor-Reynolds numbers?

line 272: what does NSTAP stands for?

Line 285/286: I am still not quite sure if I have correctly understood the difference between the "true mean" and the "ground-truth". Could you please explain these two terms (or the difference) again?!

Line 339ff: I think the sentences here are all correct in terms of content and technique, but it is very hard and complicated to read these sentences without losing the flow. You put information that could also be presented in a table (although you have already a lot of tables) into one sentence and you almost have to "study" these sentences to get the content completely. I fear that many readers will not be ready for that. This is only a prominent example and you should perhaps think about it at some other places whether one can represent the content for the reader not also somewhat more simply without neglecting thereby information / precision.

Line 343: Is the phrase "second-order dissipative statistics" correct? I wonder about the word "dissipative" in this context because the statistics cannot be "dissipative" or did I misunderstand? Please comment on this, maybe I am wrong here.

Line 412: \citep => \cite (such as in line 410)

About Eq 17: It is not clear to me what the variable "y" in the equation means exactly and how I can apply it. I think a few more explanations would be helpful at this point.

Line 440ff: Such statements (or recommendations) are very useful for and should be placed or repeated at a more prominent place in the manuscript (=> summary)

Eq 23: I cannot technically follow this equation with its different underbraces, please at least double check!

**About the summary:**

I have no concerns about writing a summary in bullet points. However, I think that a pure summary in its present form could be improved with little effort by also summarizing again here the recommendations as formulated in the discussion. This would probably be a nice conclusion to the work and the reader would take away even more hints for own applications.

I am not sure if the following suggestion will work, but one could also think about discussing at the end of the manuscript (maybe in the discussion) using a measurement example from the atmosphere to exemplify at least some methods and estimate errors. This is just an idea/suggestion and does not have to be implemented at all but it would make the paper much more interesting for more experimentally oriented readers.

**Literature:**

Line 783: please check the author's list

Line 823: not sure about "grew" literature – will probably checked by the publisher; same with other pre-prints such as in line 839

A few more papers which might be of interest in this context and also might be considered:

Rod Frehlich's work about hot-wire calibration is somehow related to your work and definitvely should be included at a prominent place:
https://doi.org/10.1175/15200469(2003)060<2487:TMWTCT>2.0.CO;2

Andreas Muschinski, R. G. Frehlich, M. L. Jensen, R. Hugo, A. M. Hoff, F. Eaton, and B. B. Balsley. Fine-scale measurements of turbulence in the lower troposphere: An intercomparison between a kite- and balloon- borne and a helicopter-borne measurement system. Boundary-Layer Meteorol., 98:219–250, 2001.

See Fig 7 in H. Siebert, S. Gerashchenko, K. Lehmann, A. Gylfason, L. R. Collins, R. A. Shaw, and Z. Warhaft. Towards understanding the role of turbulence on droplets in clouds: In situ and laboratory measurements, and numerical modeling. Atmos. Res., 97(4):426–437, 10.1016/j.atmosres.2010.05.007 2010.

The last reference includes at least a rough intercomparison of direct estimates of epsilon and inertial subrange scaling methods although it is by far not as detailed as your work.

---

## Author Comment (AC1)

[Comment types: AC – author | RC – referee]

**Responses to the comments of the Anonymous Referee #1**

[RC1.1] This paper addresses the fundamental problem of estimating "local" energy losses from one-dimensional velocity measurements as a function of time, with a focus on atmospheric conditions. Such observations can be made, for example, with single- component hot-wire anemometers and are critical to a better understanding of turbulent processes in the atmosphere. Since there are different methods and approaches to determine energy dissipation rates, there is also no doubt about the need to discuss in detail the methods and the sources of error associated with the measurement which has been done in great detail in this work. Furthermore, there is also no doubt that an in-depth treatment of this topic is still missing in the literature and thus this work makes a very valuable contribution to atmospheric turbulence measurements and their accuracy. I can only congratulate the authors on this contribution and really have only one but important comment, which relates to its potential usefulness to a wider readership and should be easy to incorporate. I must also emphasize in this context that my background is rather experimental and I have not followed all derivations in detail. Therefore, as I said before, my main criticism also lies somewhat in the potential applicability of the material for experimenters. The discussions are for the most part very theoretical which is certainly absolutely okay, but from my perspective the applicability of the results suffers as a result. I think, especially in the summary, more clear recommendations for action could be formulated as to what is the most robust method for determining dissipation rates under given circumstances. My fear is that in future publications on experimental determinations of dissipation, this manuscript will be cited but rather generally and without direct application and that would be a pity because this manuscript definitely has the potential to serve as a reference for future measurements of dissipation. After consideration of my comments, I unreservedly recommend the paper for publication. When I ask here for "major revisions", this does not refer to the scientific content but to the request to revise the paper for better applicability and partly also readability.

[AC1.1] We thank you for the positive and detailed evaluation of our work. We are pleased that you share the same point of view as we do, namely that a detailed evaluation of the subject is necessary. We also understand that the length of the results and data presented could hinder the use of the results as also pointed out by Referee #2. Following your detailed and useful suggestions, we have addressed this and other shortcomings, which we believe has greatly improved the manuscript. We have responded to all your comments point by point, with the your comments in blue and our responses in black with quotes from the revised paper in red. We also corrected minor issues here and there and polished some parts of the manuscript without changing the basic scientific content to improve the readability of the paper.

[RC1.2] The derivations and subsequent discussions of the various error contributions in the determination of the dissipation rate are described in great detail and quite formally - this is also absolutely fine and understandable, although the abundance of details reminds me partly more of a dissertation than a manuscript for AMT; however, you have chosen a meteorologically oriented, metrological journal with a corresponding audience. For this reason, I wonder how you can possibly get even more use for atmospheric measurements from this really great material. You already shifted a lot of details into the appendix which is highly appreciated but I feel that at several places there is still too much detail not necessary for estimating the uncertainty of dissipation. One prominent example in this context is the description of the active grid of your wind tunnel (in particular Eq. 12), this is very technical and does not contribute to the main topic at all - a reference would be more than sufficient. The same with some tables such as Tab 2 for example - do I really need all the details of the DNS? I think too much detail can easily distract from the main topic and slow down the reading flow, and I suggest going through the manuscript carefully and - where possible - shortening it some more (although I know from own experience how hard and difficult shortening can be...).

[AC1.2] Agreed. We were aware of the fact that the manuscript is too long and moved some of the material into the Appendices as much as possible, as you pointed out. We feel that Table 2 is needed because it contains some essential information that is relevant to the rest of the paper, e.g., when we compare results of different DNS cases. However, we found that it could do with less detail by moving some of the repetitive elements to the table caption. We have tried to simplify the text further in several places based on your suggestions (e.g., by deleting equation 12 or adding a table in response to RC1.19).

[RC1.3] A second way to make the material more valuable to a wider audience would be to include some sort of recommendation in the summary section. Such a brief discussion would be very helpful for application-oriented colleagues.

[AC1.3] This is a great idea, thank you very much. We have added a new section titled "Practical Guidelines" to help others plan their experimental setups and evaluate the uncertainties associated with estimating the energy distribution rate.

[RC1.4] Abstract: I think it is not common to define abbreviations in the abstract if they are not used again in the abstract itself - I suggest avoiding it.

[AC1.4] We have removed the abbreviations.

[RC1.5] The introduction is general nicely and clearly written with only a few places where I suggest some more details and information: Line 29ff: Maybe a short explanation what exactly is meant by "instantaneous energy dissipation field" is helpful at this place; what are the "high spatial/temporal scales" mentioned in line 29? Please specify!

[AC1.5] We have now clarified these as follow:

By "instantaneous" we here want to emphasize that $\epsilon_0$ is the energy dissipation rate at one point in space and time within the flow.

and

...is extremely difficult to measure experimentally because it requires complete knowledge of the three-dimensional velocity field with spatial/temporal resolution that can resolve scales smaller than or at least comparable to Kolmogorov scales.

[RC1.6] line 32: if the averaging is over the instantaneous energy dissipation rate, why is the index "0" missing here?

[AC1.6] Thank you for noting this, we have fixed it now (also in other places).

[RC1.7] line 36ff: can you please provide an example why the locally averaged dissipation is of importance (although I completely agree with this statement)?

[AC1.7] Yes, we have provided an example to illustrate the importance better:

For example, the local dissipation rate determines whether droplets in a cloud behave as tracer or inertial particles, which in turn can affect the probability of collision/coalescence of the droplets and thus the likelihood of precipitation initiation (e.g., see Shaw, 2003)

[RC1.8] line 49ff: this is probably true, but at least airborne turbulence observations with a sufficient high true airspeed yielding a low turbulence intensity where applying Taylor's hypothesis should be fine - right? So your comment is more related to ground-based observations or tethered systems (balloons, kites) combined with high turbulence intensities – should be mentioned.

[AC1.8] That is a good point. We have now clarified that we meant that the assumption of stationary homogeneous isotropic turbulence cannot be satisfied in general, independent of the applicability of the Taylor's hypothesis:

However, in atmospheric flows, the assumption of *ideal* stationary homogeneous isotropic turbulence needs to be very carefully considered, as for example thermals, change in local weather conditions and of course the diurnal cycle may lead to non-stationarity and inhomogeneity.

[RC1.9] I don't understand why the strain rate tensor in Eq 3 is in capitols but in tab 1 not – please specify.

[AC1.9] The reason for the change was explained after Eq. 3. However, the discrepancy between Equation 3 and Table 1 has rendered them inconsistent. Thank you for noticing this. We have rearranged the explanations and changed Equation 3 so that it is easy to follow and consistent with Table 1:

As the velocity gradients are dominated by small-scale fluctuations, turbulent kinetic energy is dissipated into heat at small scales. Therefore, the contribution of large-scale fluctuations of the velocity is small compared to the contribution of the small scales (Pope, 2000; Elsner and Elsner, 1996). Hence, the instantaneous energy dissipation rate can be defined in terms of the velocity fluctuations only, i.e., replacing $S_{ij}$ by the fluctuation strain rate tensor $s_{ij} = (\partial u_i'(\boldsymbol{x},t)/\partial x_j + \partial u_j'(\boldsymbol{x},t)/\partial x_i)/2$ (Pope, 2000).

[RC1.10] Title of Sec 2.2.3: not sure about how the subtitle compares to the subtitle of subsection 2.2.2. The spectral method is also an "indirect method" right? So, I think the subtitles should be similar and only differ for "spectral" and "structure function"?

[AC1.10] Agreed, we changed the titles as suggested.

[RC1.11] Eq 11: maybe I missed it, but the autocorrelation function f(r) has not been defined/introduced yet – right?

[AC1.11] Thanks for noticing, we have now introduced f(r) after equation 11.

[RC1.12] Fig 1A and C: Symbols are partly poorly resolved and pixelated and therefore difficult to read.

[AC1.12] This should be a problem caused by the fact that only the PDF of the manuscript is submitted during the first submission. All illustration files are in vector format, and once they are provided to the publisher in the final stage, the quality should be significantly improved.

[RC1.13] line 261: please provide a reference for the given number range for Rl under atmospheric conditions

[AC1.13] We have added a reference.

[RC1.14] line 262: maybe for the broader audience one or two sentences should be included about the basic motivation using SF6 at this high pressure and why not working at atmospheric conditions. I think the most of the readers do not know about the advantage of SF6 and what type of gas it is.

[AC1.14] This is also a good suggestion. We have added a short explanation to emphasize the advantages of using SF6 compared to air.

[RC1.15] I'm not sure Eq 12 is necessary to understand what the experiment is about; the equation formalizes the text unnecessarily and a description in the text is more helpful and sufficient

[AC1.15] Agreed. We have removed this equation and some extra details associated with it.

[RC1.16] Line 270: Why using three setups with quite comparable Taylor-Reynolds numbers?

[AC1.16] These were only the cases where the wind tunnel flow is believed to be anisotropic. The actual range of the Taylor scale for the Reynolds number, taken from wind tunnel data, is between 147 and 5864. We have decluttered this part a bit to make it easier to see the actual range used.

[RC1.17] line 272: what does NSTAP stands for?

[AC1.17] NanoScale Thermal Anemometry Probes. We have now capitalized each word to make it clearer.

[RC1.18] Line 285/286: I am still not quite sure if I have correctly understood the difference between the "true mean" and the "ground-truth". Could you please explain these two terms (or the difference) again?!

[AC1.18] Now that you have pointed it out, we also see the confusion in defining two seemingly similar terms, but that is unintentional and a matter of poor word choice. We have made these sentences a little clearer:

Generally, there are two different errors when estimating the mean energy dissipation rate, namely the systematic errors and random errors. The latter is related to the estimation variance of the mean energy dissipation rate, i.e., the statistical scatter of the $\langle \epsilon \rangle_R$-estimates around the ground truth of the local mean energy dissipation rate defined in Eq. 4. The systematic error of the mean energy dissipation rate estimates expresses itself in a non-vanishing ensemble average of the deviations from the ground-truth, i.e., the global volume average defined in Eq. 5.

[RC1.19] Line 339ff: I think the sentences here are all correct in terms of content and technique, but it is very hard and complicated to read these sentences without losing the flow. You put information that could also be presented in a table (although you have already a lot of tables) into one sentence and you almost have to "study" these sentences to get the content completely. I fear that many readers will not be ready for that. This is only a prominent example and you should perhaps think

about it at some other places whether one can represent the content for the reader not also somewhat more simply without neglecting thereby information / precision.

[AC1.19] Agreed. We have now transferred the statistics in the text into a table, i.e. Table 4, and also shortened the explanations that followed in order to focus on the most important points instead of specifying all the fine details.

[RC1.20]Line 343: Is the phrase "second-order dissipative statistics" correct? I wonder about the word "dissipative" in this context because the statistics cannot be "dissipative" or did I misunderstand? Please comment on this, maybe I am wrong here.

[AC1.20] We see your point. To avoid confusion we rephrased it to (dissipation-range) second-order statistics.

[RC1.21] Line 412: citep => cite (such as in line 410)

[AC1.21] Thank you, we have fixed this and some other similar ones.

[RC1.22] About Eq 17: It is not clear to me what the variable "y" in the equation means exactly and how I can apply it. I think a few more explanations would be helpful at this point.

[AC1.22] It is the differential variable, i.e. the variable with respect to which the integral is taken. We have moved the "dy" to the end of RHS to make this clearer.

[RC1.23] Line 440ff: Such statements (or recommendations) are very useful for and should be placed or repeated at a more prominent place in the manuscript (=> summary)

[AC1.23] Agreed, we have refereed to this point in our practical guidelines.

[RC1.24] Eq 23: I cannot technically follow this equation with its different underbraces, please at least double check!

[AC1.24] We defined the underbraced terms after the equation to make it clearer.

[RC1.25] About the summary: I have no concerns about writing a summary in bullet points. However, I think that a pure summary in its present form could be improved with little effort by also summarizing again here the recommendations as formulated in the discussion. This would probably be a nice conclusion to the work and the reader would take away even more hints for own applications.

[AC1.25] As mentioned in response to comment RC1.3, we now have a non-technical summary in the newly added "Practical guidelines" section.

[RC1.26] I am not sure if the following suggestion will work, but one could also think about discussing at the end of the manuscript (maybe in the discussion) using a measurement example from the atmosphere to exemplify at least some methods and estimate errors. This is just an idea/suggestion and does not have to be implemented at all but it would make the paper much more interesting for more experimentally oriented readers.

[AC1.26] In fact, this was one of our initial ideas, so we understand the added benefit of doing this. However, after we had written the text and it had developed to this length, we decided not to add the experimental data, as this would have lengthened the text by another page or two, especially since Figure 8 provides this information anyway, albeit from controlled wind tunnel data.

[RC1.27] Line 783: please check the author's list

[AC1.27] Thank you, we have fixed the reference.

[RC1.28] Line 823: not sure about "grew" literature – will probably checked by the publisher; same with other pre-prints such as in line 839

[AC1.28] We could not identify any particular issue to fix.

[RC1.29] A few more papers which might be of interest in this context and also might be considered: Rod Frehlich's work about hot-wire calibration is somehow related to your work and definitvely should be included at a prominent place: https://doi.org/10.1175/15200469(2003)060<2487:TMWTCT>2.0.CO;2 Andreas Muschinski, R. G. Frehlich, M. L. Jensen, R. Hugo, A. M. Hoff, F. Eaton, and B. B. Balsley. Fine-scale measurements of turbulence in the lower troposphere: An intercomparison between a kite- and balloon- borne and a helicopter-borne measurement system. Boundary-Layer Meteorol., 98:219–250, 2001.

[AC1.29] We were aware of these insightful papers and have cited them in our other publications. We have now cited them in this manuscript as well, as you suggested.

[RC1.30] See Fig 7 in H. Siebert, S. Gerashchenko, K. Lehmann, A. Gylfason, L. R. Collins, R. A. Shaw, and Z. Warhaft. Towards understanding the role of turbulence on droplets in clouds: In situ and laboratory measurements, and numerical modeling. Atmos. Res., 97(4):426–437, 10.1016/j.atmosres.2010.05.007 2010. The last reference includes at least a rough intercomparison of direct estimates of epsilon and inertial subrange scaling methods although it is by far not as detailed as your work.

[AC1.30] Thank you very much for this comment. This work is indeed relevant, and that is why we have now cited this work in the introduction along with other inter-comparison studies.

**References**

Elsner, J. and Elsner, W.: On the measurement of turbulence energy dissipation, Measurement Science and Technology, 7, 1334, 1996.

Pope, S. B.: Turbulent flows, Cambridge University Press, 2000.

Shaw, R. A.: Particle-turbulence interactions in atmospheric clouds, Annual Review of Fluid Mechanics, 35, 183–227, 2003.

---

## Author Comment (AC2)

**Responses to the comments of the Anonymous Referee #2**

[RC2.1] The manuscript investigates the potential errors in the estimation of the local turbulent kinetic energy dissipation rate. The original aspect is the use of fully resolved DNS of statistically stationary, homogeneous, isotropic turbulence to estimate the energy dissipation rate by applying to simulation data the different methodologies commonly used by experimental scientists from time-dependent single point one-dimensional velocity measurements sampled in the atmospheric boundary layer.

The DNS data are used as 'ground-truth reference' for comparing the various estimation techniques used for extracting the time-dependent energy dissipation rate from non-ideal turbulent flows and for assessing the influence of different potential causes of errors, such as (among the most relevant) the size of averaging window, the turbulence intensity, the large-scale random flow velocities, or the anemometer misalignment. The topic is extremely interesting because the turbulent kinetic energy dissipation rate is one of the most fundamental quantities in turbulence and it is crucial to accurately derive the errors associated to the different methodologies commonly used to derive it.

I read the review of the Referee #1 and globally I agree with his comments. Surely the manuscript is well written and organized and the obtained results interesting and original. Probably is 'too much' and should be simplified by cutting some details or some theoretical part, as already highlighted by the Referee #1. However, the paper is of excellent quality and I recommend to accept the paper.

[AC2.1] Many thanks for the positive review. We agree with you and Referee #1 that the amount of detail is overwhelming and may compromise the usefulness of the paper. We have streamlined the manuscript and added a new section that is essentially a non-technical summary for those who want to grasp the key and practical points of the paper quickly without diving into the details. We also corrected minor issues here and there and polished some parts of the manuscript without changing the basic scientific content to improve the readability of the paper.